# Disentangled Graph Spectral Domain Adaptation

Liang Yang[1]  Xin Chen[1]  Jiaming Zhuo[1]  Di Jin[2]  Chuan Wang[3]  Xiaochun Cao[4]  Zhen Wang[5]
Yuanfang Guo[6]

## Abstract

The distribution shifts and the scarcity of labels prevent graph learning methods, especially graph neural networks (GNNs), from generalizing across domains. Compared to Unsupervised Domain Adaptation (UDA) with embedding alignment, Unsupervised Graph Domain Adaptation (UGDA) becomes more challenging in light of the attribute and topology entanglement in the representation. Beyond embedding alignment, UGDA turns to topology alignment but is limited by the ability of the employed topology model and the estimation of pseudo labels. To alleviate this issue, this paper proposed a Disentangled Graph Spectral Domain adaptation (DGSDA) by disentangling attribute and topology alignments and directly aligning flexible graph spectral filters beyond topology. Specifically, Bernstein polynomial approximation, which mimics the behavior of the function to be approximated to a remarkable degree, is employed to capture complicated topology characteristics and avoid the expensive eigenvalue decomposition. Theoretical analysis reveals the tight GDA bound of DGSDA and the rationality of polynomial coefficient regularization. Quantitative and qualitative experiments justify the superiority of the proposed DGSDA.

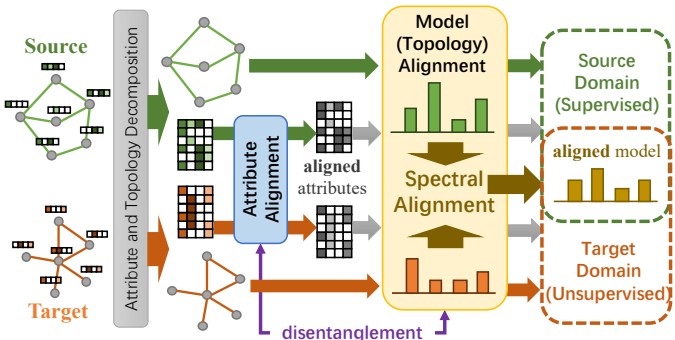

*Figure 1.* The proposed Disentangled Graph Spectral Domain Adaptation (DGSDA). The disentanglement is between attribute and model/topology alignments.

## 1. Introduction

Graphs are ubiquitous as a language for modeling complex relational data in diverse fields, ranging from social networks to traffic to the sciences (Hu et al., 2020; Li et al., 2022). Research on graphs has been conducted in many disciplines, including graph theory in mathematics (Bondy & Murty, 2008), network science in physics (Barabási, 2013), and graph representation learning in artificial intelligence (Cui et al., 2019; Guan et al., 2024; Zhuo et al., 2023; Fang et al., 2022). Unfortunately, their essential complexity makes labeling graphs more difficult and requires more domain knowledge than labeling images, text, and speech. Although the quest for the foundation model has come a long way in many fields (Bommasani et al., 2021), e.g., LM in NLP and CV (Zhao et al., 2023), the distribution shifts and scarcity of labels prevent graph learning methods, especially graph neural networks (GNNs), from generalizing across domains, and impede the design of graph foundation models (Li et al., 2024; Fan et al., 2024; Liu et al., 2024b).

Unsupervised graph learning methods have demonstrated the potential to learn transferable representations without relying on labeled data (Yang et al., 2025; Zhuo et al., 2024). However, these methods often assume a single-domain setting, overlooking cross-domain distribution shifts. Unsupervised Domain Adaptation (UDA) transfers knowledge from label-rich domains to unlabeled target domains with distribution discrepancies. Embedding alignment is

[1]Hebei Province Key Laboratory of Big Data Calculation, School of Artificial Intelligence, Hebei University of Technology, Tianjin, China [2]College of Intelligence and Computing, Tianjin University, Tianjin, China [3]School of Computer Science and Technology, Beijing JiaoTong University, Beijing, China [4]School of Cyber Science and Technology, Shenzhen Campus of Sun Yatsen University, Shenzhen, China [5]School of Artificial Intelligence, OPtics and ElectroNics (iOPEN), School of Cybersecurity, Northwestern Polytechnical University, Xi'an, China [6]School of Computer Science and Engineering, Beihang University, Beijing, China. Correspondence to: Di Jin <jindi@tju.edu.cn>.

a widely adopted methodology in UDA (Liu et al., 2022), and many alignment strategies have been proposed (Ganin et al., 2016b; Shen et al., 2018). Unsupervised Graph Domain Adaptation (UGDA) introduces the UDA problem to the graph domain (Liu et al., 2024b; Shi et al., 2024), and embedding alignment strategies are employed to handle distribution shifts (Zhang et al., 2019; Wu et al., 2020; Pang et al., 2023). However, graph data contains two different types of information, i.e., topology and node attributes, both of which may suffer from distribution shifts (Liu et al., 2024b; Shi et al., 2024; Fang et al., 2025). Thus, UGDA becomes more challenging in light of the attribute and topology entanglement in the representation compared to UDA (Ma et al., 2019).

Beyond drawing on UDA, UGDA turns to topology shift and alignment (Liu et al., 2023; 2024c), which is the problem specific to UGDA. The topology alignment needs additional topology structure models, such as Contextual Stochastic Block Models (CSBM) (Deshpande et al., 2018), to efficiently align topologies in source and target domains. Thus, the expressive ability of topology structure models is critical for knowledge transfer among graph domains. Besides, robust topology alignment often relies on accurate pseudo-label estimation in the target domain. However, this is often difficult as label prediction is the core task in UDA. In conclusion, the ability of the employed topology model and the accurate estimation of pseudo-labels hinder the quality of the topology alignment.

To alleviate this issue, this paper proposed a Disentangled Graph Spectral Domain adaptation (DGSDA) as shown in Fig. 1 by disentangling attribute and topology alignments and directly aligning complicated graph spectral filters beyond topology. Firstly, DGSDA disentangles the UGDA into attribute alignment, which has been widely investigated in UDA, and topology alignment based on the aligned node attribute. Secondly, based on the close relationship between topology and GNNs, especially spectral ones, topology alignment is converted to model alignment. The model alignment possesses the advantages of (1) end-to-end modeling, (2) parameter efficiency, and (3) benefit from a large amount of flexible GNNs. Thirdly, BernNet (He et al., 2021), which uses Bernstein polynomial approximation, is adopted, and coefficients of the polynomial are aligned between source and target domains.

The final objective function is the combination of the losses for attribute alignment, model alignment, supervised error in the source domain, and clustering regularization in the target domain. Theoretical analysis demonstrates the tight DA bound for the proposed DGSDA. The Lipschitz continuous of Bernstein polynomial on mimicking the behavior of the function to be approximated makes the spectral Lipschitz constant determined by the ground truth function to be

learned instead of the employed polynomial. Besides, the analysis also justifies the polynomial coefficient regularization in the model alignment loss.

The main contributions are summarized as follows:

- We introduce a novel UGDA pipeline by disentangling attribute and topology alignments and replacing topology alignment with model alignment.
- We propose a novel Disentangled Graph Spectral DA (DGSDA) by directly aligning spectral filters. DGSDA is end-to-end, parameter efficient, and possesses high expressive ability.
- We present the tight DA bound for our DGSDA with Bernstein polynomials and justify the alignment loss.
- We conduct experiments to show the proposed UGSDA achieves a new SOTA.

## 2. Related Work

Traditional domain adaptation approaches use intermediate representations to minimize domain discrepancy, which can be categorized into two main streams: methods that minimize pre-defined probability discrepancy metrics (Gretton et al., 2012; Zellinger et al., 2017) and those that employ adversarial learning techniques (Ganin et al., 2016a; Long et al., 2018; Tzeng et al., 2017). However, these methods are not appropriate for graph-structured data.

Recently, several approaches have been proposed to address the unique challenges of UGDA (Zhang et al., 2021; Wang et al., 2023; Shen et al., 2023; Cai et al., 2024; Huang et al., 2024). Notable methods include DANE (Zhang et al., 2019), which uses shared GCNs and a least square generative adversarial network. ACDNE (Shen et al., 2020) employs feature extractors and a domain classifier. UDAGCN (Wu et al., 2020) and AdaGCN (Dai et al., 2022) integrates graph convolution with adversarial training for graph transfer learning. CoCo (Yin et al., 2023) and DREAM (Yin et al., 2024) are designed for the classification of graph-level domain adaptation. These approaches inherit limitations from conventional domain adaptation: they predominantly address feature-level shifts while neglecting structural misalignment.

Beyond addressing feature distribution shifts, recent studies explore structural adaptation strategies for graph domain shifts. StruRW (Liu et al., 2023) develops an edge reweighting mechanism that mitigates conditional neighborhood distribution shifts across domains. Extending this paradigm, PairAlign (Liu et al., 2024c) introduces a dual adaptation framework that simultaneously recalibrates node influence through adaptive edge weighting and counteracts label distribution mismatch via classification loss reweighting. JHGDA (Shi et al., 2023) designs a multi-level pooling model to extracts hierarchical representations and compute

domain discrepancies at different levels. KBL (Bi et al., 2023) proposes an alternative paradigm via bridged-graph knowledge transfer. These approaches fundamentally couple node representation learning with structural adaptation in a joint optimization framework. This tight integration risks entangling domain-invariant patterns with topology-specific artifacts, potentially amplifying spurious correlations.

Beyond simply using GNNs as node embedding modules, emerging research systematically investigates their intrinsic properties for domain adaptation scenarios. SpecReg (You et al., 2023) establishes theoretical connections between optimal transport-based generalization bounds and GNN spectral characteristics, revealing that the Lipschitz continuity of graph filters fundamentally constrains cross-domain risk. A2GNN (Liu et al., 2024a) finds that the propagation operation plays a pivotal role and proposes a simple GNN that stacks more propagation layers on the target branch.

## 3. Preliminaries

### 3.1. Notations

A graph can be represented as $\mathcal{G} = (\mathcal{V}, \mathcal{E})$ where $\mathcal{V}$ and $\mathcal{E}$ are the sets of nodes and edges. $N = |\mathcal{V}|$ and $M = |\mathcal{E}|$ stand for the numbers of nodes and edges. The node attribute matrix, denoted by $\mathbf{X} = \{\mathbf{x}_i | v_i \in \mathcal{V}\} \in \mathbb{R}^{N \times F}$, contains attribute vector $\mathbf{x}_i$ for each node $v_i$, where $F$ represents the dimensionality of the attributes. Adjacency matrix of $\mathcal{G}$ is represented as $\mathbf{A} = [a_{ij}] \in \mathbb{R}^{N \times N}$. $a_{ij} = 1$ holds if there is an edge $e_{ij} \in \mathcal{E}$ connecting nodes $v_i$ and $v_j$, and $a_{ij} = 0$ otherwise. $\mathcal{N}(v_i)$ stands for the neighborhood of node $v_i$. $\mathbf{D} = [d_{ii}] \in \mathbb{R}^{N \times N}$ denotes the degree matrix with diagonal element $d_{ii} = \sum_{v_j \in \mathcal{N}(v_i)} a_{ij}$ as the degree of node $v_i$. $\mathbf{P} = \mathbf{D}^{\frac{1}{2}} \mathbf{A} \mathbf{D}^{\frac{1}{2}}$ is the normalized adjacency matrix. $\hat{\mathbf{L}} = \mathbf{D} - \mathbf{A}$ and $\mathbf{L} = \mathbf{I} - \mathbf{D}^{\frac{1}{2}} \mathbf{A} \mathbf{D}^{\frac{1}{2}}$ represent the Laplacian matrix and its symmetric normalized version, where $\mathbf{I}$ stands for the identity matrix. $\mathbf{Y} \in \mathbb{R}^{N \times C}$ represents the node label matrix, where $C$ denotes the number of classes.

### 3.2. Problem Definition

Given a labeled source graph $\mathcal{G}^S = (\mathcal{V}^S, \mathcal{E}^S, \mathcal{Y}^S)$ and an unlabeled target graph $\mathcal{G}^T = (\mathcal{V}^T, \mathcal{E}^T)$ with the data shift that $\mathbb{P}(\mathcal{G}^S) \neq \mathbb{P}(\mathcal{G}^T)$ or equivalently $\mathbb{P}(\mathbf{A}^S, \mathbf{X}^S | \mathbf{Y}^S) \neq \mathbb{P}(\mathbf{A}^T, \mathbf{X}^T | \mathbf{Y}^T)$. Superscripts $\cdot^S$ and $\cdot^T$ stand for the source and target domains, respectively. Domain indicator can be placed on $\mathbb{P}$ for simplicity, e.g. $\mathbb{P}^U(\mathbf{A}, \mathbf{X} | \mathbf{Y}) = \mathbb{P}(\mathbf{A}^U, \mathbf{X}^U | \mathbf{Y})$ for $U \in \{S, T\}$. Unsupervised Graph Domain Adaptation (UGDA) is to seek a model $g : \mathcal{G}^T \to \mathcal{Y}^T$ which can be generalized to tasks on unlabeled target graph $\mathcal{G}^T$. Model $g : \mathcal{G} \to \mathcal{Y}$ consists of a feature extractor $h : \mathcal{G} \to \mathcal{H}$ and a classifier $f : \mathcal{H} \to \mathcal{Y}$. Here, the node-level classification task is mainly considered. Both the labeled source graph $\mathcal{G}^S$ and the unlabeled target graph $\mathcal{G}^S$

are employed to train the model $g : \mathcal{G}^T \to \mathcal{Y}^T$.

### 3.3. Spectral Graph Neural Networks

Let $\mathbf{L} = \mathbf{U} \mathbf{\Lambda} \mathbf{U}^T$ denote the eigen-decomposition of the symmetric normalized Laplacian matrix $\mathbf{L}$, where $\mathbf{U}$ is the matrix of eigenvectors and $\mathbf{\Lambda} = diag[\lambda_1, ..., \lambda_N]$ is the matrix of eigenvalues. The spectral filter on the graph is

$$h(\mathbf{L})\mathbf{x} = \mathbf{U}h(\mathbf{\Lambda})\mathbf{U}^T\mathbf{x} = \mathbf{U}diag[h(\lambda_1), ..., h(\lambda_N)]\mathbf{U}^T\mathbf{x}, \tag{1}$$

where $\lambda_i \in [0, 2]$ for $i = 1, ..., N$ and $\mathbf{x} \in \mathbb{R}^N$ stands for graph signal. Spectral graph neural networks aim to design and learn the mapping function $h(\mathbf{L})$, or equivalently $h(\lambda)$. Different polynomial approximations are employed to fit $h(\lambda)$, such as Chebyshev polynomials (Defferrard et al., 2016; Kipf & Welling, 2017; He et al., 2022), Bernstein polynomials (He et al., 2021), and Jacobi polynomial (Wang & Zhang, 2022). Specformer (Bo et al., 2023) considers the set relationships between eigenvalues with Transformer.

## 4. Method

This section presents the Disentangled Graph Spectral Domain Adaptation (DGSDA). Sections 4.1 and 4.2 elaborate two components Distribution Shift Disentanglement and Graph Spectral Domain Adaptation, respectively. Section 4.3 provides the overall objective function and algorithm followed by the theoretical justification as in Section 4.4.

### 4.1. Distribution Shift Disentanglement

Unsupervised Graph Domain Adaptation (UGDA) aims to align the embedding condition distributions $\mathbb{P}^S(\mathbf{H} | \mathbf{Y}) = \mathbb{P}^T(\mathbf{H} | \mathbf{Y})$ to deal with the data shift $\mathbb{P}^S(\mathbf{A}, \mathbf{X} | \mathbf{Y}) \neq \mathbb{P}^T(\mathbf{A}, \mathbf{X} | \mathbf{Y})$ with the feature extractor $h : \mathcal{G} \to \mathcal{H}$. Since $\mathbb{P}(\mathbf{A}, \mathbf{X} | \mathbf{Y}) = \mathbb{P}(\mathbf{X} | \mathbf{Y})\mathbb{P}(\mathbf{A} | \mathbf{X}, \mathbf{Y})$, the graph DA process can be disentangled into two steps: node attribute alignment w.r.t. $\mathbb{P}(\mathbf{X} | \mathbf{Y})$ and topology alignment w.r.t. $\mathbb{P}(\mathbf{A} | \mathbf{X}, \mathbf{Y})$.

**Node attribute alignment.** Fortunately, there is much progress in the non-graph domain adaptation (Liu et al., 2022). They achieve $\mathbb{P}^S(\mathbf{H}_X | \mathbf{Y}) = \mathbb{P}^T(\mathbf{H}_X | \mathbf{Y})$ under scenarios of $\mathbb{P}^S(\mathbf{X} | \mathbf{Y}) \neq \mathbb{P}^T(\mathbf{X} | \mathbf{Y})$ with $h_X : \mathcal{X} \to \mathcal{H}$, where $\mathbf{X}$ stands for the collection of i.i.d. data and $\mathbf{H}_X$ for its representation. Thus, the node attribute can be aligned.

**Topology alignment.** Therefore, the graph data shift can be simplified from $\mathbb{P}^S(\mathbf{A}, \mathbf{X} | \mathbf{Y}) \neq \mathbb{P}^T(\mathbf{A}, \mathbf{X} | \mathbf{Y})$ to $\mathbb{P}^S(\mathbf{A} | \mathbf{X}, \mathbf{Y}) \neq \mathbb{P}^T(\mathbf{A} | \mathbf{X}, \mathbf{Y})$. Since the node attribute has been aligned in embedding space, node attribute divergence can be ignored and topology alignment can be further converted to the data shift scenarios of

$$\mathbb{P}^S(\mathbf{A} | \mathbf{H}_X, \mathbf{Y}) \neq \mathbb{P}^T(\mathbf{A} | \mathbf{H}_X, \mathbf{Y}). \tag{2}$$

Section 4.2 proposes a flexible topology alignment scheme

with aligned attribute embedding.

Benefiting from the above graph distribution shift disentanglement, the overall framework of the proposed Disentangled Graph Spectral Domain Adaptation (DGSDA) is shown in Fig. 1. Note that DGSDA does not require additional final embedding alignment after topology alignment. Thus, it overcomes the drawbacks of methods using entangled node embedding alignment.

## 4.2. Graph Spectral Domain Adaptation

This section proposes a novel Graph Spectral Domain Adaptation (GSDA) algorithm to achieve $\mathbb{P}^S(\mathbf{H}|\mathbf{Y}) = \mathbb{P}^T(\mathbf{H}|\mathbf{Y})$ under data shift scenarios of $\mathbb{P}^S(\mathbf{A}|\mathbf{H}_X, \mathbf{Y}) \neq \mathbb{P}^T(\mathbf{A}|\mathbf{H}_X, \mathbf{Y})$. The following two subsections show why and how to use GNNs alignment instead of topology one.

### 4.2.1. GNNs ALIGNMENT

Note that the relationship between i.i.d. date feature $\mathbf{x}$ and feature extractor $h_X : \mathcal{X} \to \mathcal{H}$ is very different from that between graph topology $\mathbf{A}$ and graph learning function $h : \mathcal{G} \to \mathcal{H}$. On the one hand, the feature of i.i.d. data $\mathbf{x}$ only acts as the input to a model $h_X$. On the other hand, graph topology, represented as adjacency matrix $\mathbf{A}$, is closely related to the graph learning scheme, such as the propagation scheme in GNNs. Taking spectral GNN as an example, graph topology determines the spectral space to filter data with the eigenvectors of its Laplacian matrix as $\mathbf{U}$ in Eq. (1). Therefore, different from existing methods (Liu et al., 2023; 2024c), which perform topology structures alignment and apply GNNs on the aligned topology, this paper directly aligns the parameterized GNNs across divergent domains shown in Eq. (2):

$$GNN_\theta^S(\mathbf{A}, \mathbf{H}_X) \overset{align}{\Longleftrightarrow} GNN_\theta^T(\mathbf{A}, \mathbf{H}_X), \qquad (3)$$

where $\theta$ is the parameters related to the graph topology in GNN. The alignment of GNN is equivalent to the parameter alignment. It possesses the following three advantages:

- End-to-end model alignment is optimal compared to the topology alignment. Model alignment is equivalent to jointly aligning the topology and selecting proper GNNs. On the contrary, topology alignment also requires choosing additional GNNs.
- Model alignment, i.e., parameter alignment, may be efficient. Compared to the number of edges, which need to be aligned and adjusted, that of model parameters is often much smaller and independent of the graph size.
- Compared to topology structure models, a larger number of GNNs exist. There are many flexible GNNs designed for different kinds of graphs ranging from homophilous to heterophilic ones. On the contrary, the topology structure model is rare.

### 4.2.2. SPECTRAL FILTER ALIGNMENT

Benefiting from the above advantages, a highly expressive GNN is required to act as the backbone for GNN alignment. Here, spectral GNN is employed for its ability to capture and model diverse characteristics of graph signals, such as low-passing, high-passing, and bandit-passing. Unfortunately, as shown in Eq. (1), vanilla spectral GNNs are computationally expensive due to the eigenvalue decomposition of the Laplacian matrix. Here, BernNet (He et al., 2021) is adopted for its simplicity, efficiency, and theoretical support for learning arbitrary graph spectral filters.

BernNet implements $h(\lambda)$ in Eq. (1) with $K$-order Bernstein polynomial approximation on $t \in [0, 1]$ as

$$h_K(t) := \sum_{k=0}^{K} \theta_k \cdot b_k^K(t) = \sum_{k=0}^{K} f\left(\frac{k}{K}\right) \binom{K}{k}(1-t)^{K-k}t^k,$$
(4)

where $b_k^K(t) = \binom{K}{k}(1-t)^{K-k}t^k$ is the $k$-th Bernstein base, and $\theta_k = f(\frac{k}{K})$ is the function value at $k/K$, which acts as the coefficient of $b_k^K(t)$. Thus, by deflating the input to $[0, 1]$, the spectral GNN for signal $\mathbf{x}$ in Eq. (1) becomes

$$\mathbf{z} = B_K(\mathbf{A})\mathbf{x} = \mathbf{U}diag[h_K(\lambda_1/2), ..., h_K(\lambda_n/2)]\mathbf{U}^\top \mathbf{x}$$
$$= \sum_{k=0}^{K} \theta_k \frac{1}{2^K}\binom{K}{k}(2\mathbf{I} - \mathbf{L})^{K-k}\mathbf{L}^k\mathbf{x}, \ (5)$$

where $\mathbf{x}$ represents a general graph signal. Since the model alignment is based on the aligned node attributes embedding $\mathbf{H}_X$ as shown in Eq. (2), the graph signal in the proposed DGSDA is the aligned node attribute, i.e., the row of $\mathbf{H}_X$.

It is proved that BernNet possesses many good properties (He et al., 2021). Firstly, for an arbitrary continuous filter function $h : [0, 2] \to [0, 1]$, the $\mathbf{z}$ in Eq. (5) can approximate $h(\mathbf{L})\mathbf{x}$ as $K \to \infty$. Secondly, BernNet does NOT need expensive eigenvalue decomposition and is thus efficient. Thirdly, BernNet exactly realizes existing filters, which are commonly used in GNNs by specifying $\theta_k$'s, such as Linear/Impulse low-pass filters, Linear/Impulse high-pass filters, and Impulse band-pass filters. Intuitively, the bases $(2\mathbf{I} - \mathbf{L}) = \mathbf{I} + \mathbf{A}$ and $\mathbf{L}$ correspond to smoothing and sharpening operations, respectively.

According to Eq. (3), the $h_K(t)$'s in Eq. (4) from source and target should be aligned. To this end, its parameters $\theta_k$'s, which denote the responses to different frequencies, need to be aligned as the following loss term.

$$\mathcal{L}_{align} = \sum_{k=0}^{K} \left|\theta_k^S - \theta_k^T\right| + \sum_{k=0}^{K} \left(\left|\theta_k^S\right| + \left|\theta_k^T\right|\right), \qquad (6)$$

where $\theta_k^S$ and $\theta_k^T$ are the coefficients for source and target domains, respectively. The second term is to regularize the model parameters for adaptation as justified in Section 4.4.

## 4.3. Objective Function and Algorithm

As shown in Fig. 1, the overall Disentangled Graph Spectral Domain Adaptation (DGSDA) consists of four components: source domain encoder, target domain encoder, node attribute alignment, and model alignment. Thus, the overall objective function is as follows

$$\mathcal{L} = \mathcal{L}_{source} + \alpha \mathcal{L}_{align} + \beta \mathcal{L}_{mmd} + \gamma \mathcal{L}_{target}, \quad (7)$$

where $\mathcal{L}_{source}$ (Eq. (11)), $\mathcal{L}_{align}$ (Eq. (6)), $\mathcal{L}_{mmd}$ (Eq. (12)) and $\mathcal{L}_{target}$ (Eq. (13)) correspond to source domain classification loss, spectral alignment loss, maximum mean discrepancy loss, and target domain unsupervised loss. $\alpha$, $\beta$, and $\gamma$ are hyper-parameters to balance these terms. Refer to the Appendix A for a detailed description of each item in the formula.

## 4.4. Theoretical Analysis

This section provides the domain adaptation (DA) bound of the proposed DGSDA by following the DA bound framework for graph-structured data in (You et al., 2023), which extends the DA bound for i.i.d. data in (Redko et al., 2017). For clarity, the definition of Lipschitz continuous is as follows.

**Definition 4.1.** Given two metric spaces $(\mathcal{X}, d_X)$ and $(\mathcal{Y}, d_Y)$, where $dX$ denotes the metric on the set $\mathcal{X}$ and $dY$ is the metric on set $\mathcal{Y}$, a function $f : \mathcal{X} \to \mathcal{Y}$ is called Lipschitz continuous of $\mu$ order, denoted as $f \in LIP_{C,\mu}$, if there exists a real constant $C \geq 0$ and $0 < \mu \leq 1$ such that, for all $x_1$ and $x_2$ in $\mathcal{X}$,

$$d_Y(f(x_1), f(x_2)) \leq C d_X(x_1, x_2)^{\mu}. \quad (8)$$

Any such C is referred to as a Lipschitz constant for the function $f$, and $f$ may also be referred to as $C$-Lipschitz for the case of the order $\mu = 1$.

With the definition of Lipschitz continuous, the DA bound for graph data can be generally expressed as follows.

**Theorem 4.2.** *(You et al., 2023) Let's assume that the learned discriminator $f$ is $C_f$-Lipschitz continuous where the Lipschitz norm $\|f\|_{Lip} = \max_{Z_1, Z_2} \frac{|f(X_1) - f(Z_2)|}{\rho(Z_1, Z_2)} = C_f$ holds for some distance function $\rho$, and the graph feature extractor $h$ (also referred to as GNN) is $C_h$-Lipschitz that $\|h\|_{Lip} = \max_{G_1, G_2} \frac{\|h(G_1) - h(G_2)\|_2}{\eta(G_1, G_2)} = C_h$ for some graph distance measure $\eta$. Let $\mathcal{F} := \{g : \mathcal{G} \to \mathcal{Y}\}$ be the set of bounded real-valued functions with the pseudo-dimension $Pdim(\mathcal{F}) = d$ that $g = f \circ h \in \mathcal{F}$, with probability at least $1 - \delta$ the following inequality holds:*

$$\begin{aligned}
\epsilon^T(g, \hat{g}) \quad \leq \quad & \hat{\epsilon}^S(g, \hat{g}) \\
+ \quad & \sqrt{\frac{4d}{N^S} \ln\left(\frac{eN^S}{d}\right) + \frac{1}{N^S} \ln\left(\frac{1}{\delta}\right)} \\
+ \quad & 2C_f C_h W_1\left(\mathbb{P}^S(G), \mathbb{P}^T(G)\right) + \omega, \quad (9)
\end{aligned}$$

*where the (empirical) source and target risks are $\hat{\epsilon}^S(g, \hat{g}) = \frac{1}{N^S} \sum_{n=1}^{N^S} |g(G_n) - \hat{g}(G_n)|$ and $\epsilon^T(g, \hat{g}) = \mathbb{E}_{\mathbb{P}^T(G)}\{g(G) - \hat{g}(G)\}$, respectively, where $\hat{g} : \mathcal{G} \to \mathcal{Y}$ is the labeling function for graphs and*

$$\omega = \min_{\|f\|_{Lip} \leq C_f, \|h\|_{Lip} \leq C_h} \left\{\epsilon^T(g, \hat{g}) + \hat{\epsilon}^S(g, \hat{g})\right\}. \quad (10)$$

Unfortunately, it isn't easy to instantiate the GNN Lipschitz constant, since the distance metric $\eta(G_1, G_2)$ often requires computationally expensive graph matching. As in (You et al., 2023), the numerator, i.e. $\|g(G_1) - g(G_2)\|_2$, which is related to the GNN stability, is estimated. Recall that $K$-order Bernstein polynomial in Eq. (5) is employed as the graph encoder. One important property of Bernstein polynomials is that they mimic the behavior of the function to be approximated to a remarkable degree. The following theorem formally demonstrates this attractive property.

**Theorem 4.3.** *if $f \in LIP_{C,\mu}$, then its K-order Bernstein polynomial approximation $h_K(t)$ for all $K \geq 1$ defined in Eq. (4) with $\theta_k = f(\frac{k}{K})$ belong to $LIP_{C,\mu}$ also.*

The proof is given in Appendix B.1. With the above theorem, $\|g(G_1) - g(G_2)\|_2$ can be estimated as follows. Given $\forall G_1, G_2$ with size $N$ and $\mathbf{L}_1 = \mathbf{U}_1 \mathbf{\Lambda}_1 \mathbf{U}_1^\top$, $\mathbf{L}_2 = \mathbf{U}_2 \mathbf{\Lambda}_2 \mathbf{U}_2^\top$, the eigenvalue decomposition for Laplacian matrices $\mathbf{L}_1$ and $\mathbf{L}_2$ that $\mathbf{\Lambda}_1 = \text{diag}([\lambda_{1,1}, \ldots, \lambda_{1,N}])$, $\mathbf{\Lambda}_2 = \text{diag}([\lambda_{2,1}, \ldots, \lambda_{2,N}])$ with eigenvalues sorted in the descending order. The proposed DGSDA is constructed by composing a graph filter $B_K(\mathbf{A})$ in Eq. (5) and non-linear mapping that $g(G_1) = \sigma(B_K(\mathbf{A}_1)\mathbf{X}_1\mathbf{W}) = \sigma\left(\mathbf{U}_1 h_K(\mathbf{\Lambda}_1)\mathbf{U}_1^\top \mathbf{X}_1 \mathbf{W}\right)$ where $h_K = \sum_{k=0}^{K} \theta_k \cdot b_k^K(t)$ is the polynomial function in Eq. (4) that $B_K(\mathbf{A}_1) = \sum_{k=0}^{K} \theta_k \frac{1}{2^K}\binom{K}{k}(2\mathbf{I} - \mathbf{L}_1)^{K-k}\mathbf{L}_1^k$, $\mathbf{W} \in \mathbb{R}^{D \times D'}$ is the learnable weight matrix, and the pointwise nonlinearity holds as $|\sigma(b) - \sigma(a)| \leq |b - a|, \forall a, b \in \mathbb{R}$.

**Theorem 4.4.** *Suppose the Bernstein polynomial $h_K$ approximate the ground truth one $\bar{f}$ with $\theta_k = \bar{f}(\frac{k}{K})$ and $\|\mathbf{X}\|_{\text{op}} \leq 1$ and $\|\mathbf{W}\|_{\text{op}} \leq 1$ where $\|\cdot\|_{\text{op}}$ stands for operator norm, the following inequality holds:*

$$\begin{aligned}
\|g(G_1) &- g(G_2)\|_2 \\
&\leq C_\lambda \left(1 + \tau\sqrt{N}\right) \left\|\mathbf{A}_1 - \mathbf{P}^* \mathbf{A}_2 \mathbf{P}^{*\top}\right\|_{\text{F}} \\
&+ \mathcal{O}\left(\left\|\mathbf{A}_1 - \mathbf{P}^* \mathbf{A}_2 \mathbf{P}^{*\top}\right\|_{\text{F}}^2\right) \\
&+ \max\{|h_K(\mathbf{\Lambda}_2)|\}\left\|\mathbf{X}_1 - \mathbf{P}^*\mathbf{X}_2\right\|_{\text{F}},
\end{aligned}$$

*where $\tau = (\|\mathbf{U}_1 - \mathbf{U}_2\|_{\text{F}} + 1)^2 - 1$ stands for the eigenvector misalignment which can be bounded, $\mathbf{P}^* = \text{argmin}_{\mathbf{P} \in \Pi}\{\|\mathbf{X}_1 - \mathbf{P}\mathbf{X}_2\|_{\text{F}} + \|\mathbf{A}_1 - \mathbf{P}\mathbf{A}_2\mathbf{P}^{*\top}\|_{\text{F}}\}$, $\Pi$ is the set of permutation matrices,*

$\mathcal{O}\left(\left\|\mathbf{A}_1 - \mathbf{P}^*\mathbf{A}_2\mathbf{P}^{*\top}\right\|_{\mathrm{F}}^2\right)$ *is the remainder term with bounded multipliers, and $C_\lambda$ is the Lipschitz constant of $\bar{f}$ that $\forall \lambda_i, \lambda_j, \left|\bar{f}(\lambda_i) - \bar{f}(\lambda_j)\right| \leq C_\lambda |\lambda_i - \lambda_j|.$*

The proof is given in Appendix B.2. Note that Theorem 4.4 is different from the Lemma 1 in (You et al., 2023). In Lemma 1 in (You et al., 2023), the Lipschitz constant $C_\lambda$ is determined by the basic polynomial function, i.e., $\forall \lambda_i, \lambda_j, |B_K(\lambda_i) - B_K(\lambda_j)| \leq C_\lambda |\lambda_i - \lambda_j|$ where $B_K(\lambda) = \sum_{k=0}^{\infty} s_k \lambda^k$ is the common polynomials. In Theorem 4.4, the Lipschitz constant $C_\lambda$ is determined by the ground truth function $\bar{f}$, since the attractive property of Bernstein polynomial in Theorem 4.3.

**Theorem 4.5.** *Let's define the matching distance between $G_1, G_2$ as $\eta(G_1, G_2) = \min_{\mathbf{P} \in \Pi} \left\{\|\mathbf{X}_1 - \mathbf{P}\mathbf{X}_2\|_{\mathrm{F}} + \left\|\mathbf{A}_1 - \mathbf{P}\mathbf{A}_2\mathbf{P}^{\top}\right\|_{\mathrm{F}}\right\}.$ Suppose that the edge perturbation is bounded that $\forall G_1, G_2, \left\|\mathbf{A}_1 - \mathbf{P}^*\mathbf{A}_2\mathbf{P}^{*\top}\right\|_{\mathrm{F}} \leq \varepsilon$ with the optimal permutation $\mathbf{P}^*$, and there exists an eigenvalue $\lambda^* \in \mathbb{R}$ to achieve the maximum $|h_K(\lambda^*)| < \infty$. The Lipschitz constant of the proposed DGSDA can be estimated as*

$$C_f = \max\left\{C_\lambda K_1 + \varepsilon K_2, |h_K(\lambda^*)|\right\},$$

*where $K_1, K_2$ is the supremes of $\left(1 + \tau\sqrt{N}\right)$ and the remainder multiplier in Theorem 4.4.*

The proof is given in Appendix B.3. According to Eq. (9) the gap between the target and source errors is bounded with two terms:

- Term $W_1\left(\mathbb{P}^S(G), \mathbb{P}^T(G)\right)$ which captures the distribution divergence between the source and target multiplied by Lipschitz constants $C_f$;
- Term $\omega$ in Eq. (10) which models the discriminative capability of model to capture invariant knowledge restricted by Lipschitz constants.

Thus, varying the Lipschitz constant may balance between domain-divergence and discriminability to vary the DA bound. To tighten the DA bound in Eq. (9) can be implemented by regularizing the Lipschitz constant $C_f$. Recall that $C_\lambda$ is determined by the ground truth function to approximate instead of the polynomial, and thus $C_\lambda$ is fixed. Therefore, only $|h_K(\lambda^*)|$ need to be regularized. To this end, the absolute values of Bernstein polynomial coefficients, i.e., $|\theta_k|$'s are regularized as shown in the second terms in Eq. (6).

## 5. Experiments

### 5.1. Experimental Setup

**Datasets.** The experiment utilizes three types of benchmark datasets: Citation networks (ArnetMiner: ACMv9 (A),

Citationv1 (C) and DBLPv7 (D)), social interactions (Blog-Catalog and Twitch-DE/EN), and transportation systems (Airport: Brazil (B), Europe (E) and USA (U)). Refer to Appendix C for dataset details.

**Baselines.** The baselines for comparison can be divided into three categories. (1) Source-only methods, including vanilla GCN (Kipf & Welling, 2017) and GAT (Velickovic et al., 2018). These methods are trained only on the source graph and directly applied to target graph for evaluation. (2) GDA methods using node embedding, containing DANE (Zhang et al., 2019), UDAGCN (Wu et al., 2020) and AdaCGN (Dai et al., 2022). These methods use node embedding to solve the graph domain adaptation problem. (3) GDA methods tailored for graph structure shift, including StruRw (Liu et al., 2023), JHGDA (Shi et al., 2023), KBL (Bi et al., 2023) and PairAlign (Liu et al., 2024c). (4) Graph domain method tailored for propagation: A2GNN (Liu et al., 2024a). These models are analyzed in the Related works.

**Configurations.** For reproducibility, the detailed settings of the experiments are described below. All experiments are performed on Nvidia GeForce RTX 3090 (24GB). Our proposed model DGSDA[1] is implemented with PyTorch (Paszke et al., 2017) and PyTorch Geometric library (Fey & Lenssen, 2019). To be fair, we use the source code provided by the authors for each baseline and fine-tune the hyperparameters to achieve optimal values. In all the experiments, we use the Adam optimizer. We run all models five times on each dataset, and the mean accuracy is used as the metric.

**Hyperparameters.** For hyperparameter settings, The node representation dimension is selected from {128, 256}. The learning rate is tuned from {0.01, 0.005, 0.001, 0.0005}, weight decay is tuned from {0.0005, 0.005, 0.01} and K is selected from {5, 8, 10, 15}.

### 5.2. Result Analysis

**Citation Network.** The experimental results on citation networks are presented in Tab. 1. From these results, two key observations can be made. First, models tailored for graph structure shifts consistently outperform models that align only node representations on all datasets. This indicates the significance of graph structure in graph domain adaptation, which fits the observations that the cross-domain differences often manifest in both citation patterns (structure) and textual features (attribute). More importantly, the proposed DGSDA achieves superior performances over baseline models, particularly, models tailored for graph structure shifts, across all citation networks. To be specific, DGSDA achieves performance improvements of 6.33% over the second-best baseline JHGDA on the (D→C) datasets. This reveals that the decoupled modeling allows mutually

---

[1] Our code is available at https://github.com/Hechriver/DGSDA

*Table 1.* Node classification performance on citation network. The metric is mean accuracy (%) and standard deviation. The best and the second best results are highlighted in bold and underlined, respectively.

| METHOD | A → C | C → A | A → D | D → A | C → D | D → C |
|---|---|---|---|---|---|---|
| GCN | 76.13±0.51 | 68.52±0.33 | 68.80±0.22 | 62.42±0.40 | 72.97±0.17 | 72.53±0.25 |
| GAT | 74.45±0.41 | 68.15±0.41 | 69.43±0.76 | 62.11±0.70 | 72.52±0.38 | 71.65±0.62 |
| DANE | 65.80±0.13 | 65.39±0.43 | 63.98±0.28 | 60.28±0.78 | 65.98±0.36 | 71.29±0.44 |
| UDAGCN | 72.06±0.15 | 70.28±0.17 | 70.93±0.53 | 63.42±0.67 | 73.03±0.19 | 71.15±0.11 |
| ADAGCN | 74.28±0.28 | 67.89±0.34 | 67.83±0.85 | 60.52±0.82 | 72.83±0.89 | 71.98±0.64 |
| STRURW | 77.35±0.16 | 71.23±0.16 | 73.51±0.28 | 65.19±0.16 | 74.33±0.21 | 74.28±0.16 |
| JHGDA | 77.28±0.53 | 73.72±0.28 | 73.13±0.28 | 69.80±0.21 | 74.25±0.43 | 76.59±0.34 |
| KBL | 78.21±0.36 | 71.49±0.16 | 69.28±0.37 | 63.45±0.28 | 74.96±0.25 | 73.92±0.14 |
| PAIRALIGN | 73.76±0.13 | 70.25±0.27 | 69.18±0.17 | 62.88±0.67 | 72.59±0.19 | 72.35±0.36 |
| A2GNN | 80.68±0.98 | 75.12±0.54 | 76.13±0.39 | 73.48±0.38 | 76.15±0.73 | 79.89±0.36 |
| DGSDA | **83.57±0.22** | **75.54±0.28** | **76.90±0.51** | **74.07±0.56** | **78.38±0.28** | **82.92±0.15** |

*Table 2.* Node classification performance on social network. The metric is mean accuracy (%) and standard deviation. The best and the second best results are in bold and underlined, respectively.

| METHOD | DE→EN | EN→DE | B1→B2 | B2→B1 |
|---|---|---|---|---|
| GCN | 54.42±0.89 | 61.00±0.30 | 40.66±4.98 | 40.15±3.46 |
| GAT | 54.56±0.01 | 59.32±0.28 | 28.55±5.67 | 24.48±5.57 |
| DANE | 51.84±0.16 | 51.69±0.36 | 32.17±2.08 | 31.86±0.33 |
| UDAGCN | 58.23±0.61 | 58.61±0.28 | 33.51±2.38 | 34.16±2.18 |
| ADAGCN | 57.18±0.89 | 58.01±0.67 | 41.03±1.23 | 37.69±4.18 |
| STRURW | 58.27±0.18 | 62.57±0.39 | 48.19±0.95 | 41.53±0.96 |
| JHGDA | 56.50±0.41 | **63.17±0.11** | 20.89±2.26 | 23.86±4.18 |
| KBL | 58.89±0.38 | 60.54±0.16 | 35.63±2.89 | 34.89±1.28 |
| PAIRALIGN | 56.56±0.28 | 58.75±2.27 | 39.83±8.61 | 45.18±3.67 |
| A2GNN | 56.52±0.38 | 61.51±0.83 | 24.58±2.53 | 33.16±2.18 |
| DGSDA | **59.81±0.18** | 62.86±0.22 | **65.64±1.98** | **63.99±4.34** |

non-disruptive alignment processes for citation networks, where attributes and structures represent complementary semantic hierarchies. The disentanglement strategy inherently respects their distinct roles in cross-domain transfer.

**Social Network.** Similar conclusions regarding performance advantages can be drawn from the experiment results on social networks. It can be observed from Tab. 2 that the proposed DGSDA has performance advantages on three of the four datasets, which demonstrates its effectiveness and universality. It is worth noting that on the Blog dataset, DGSDA achieves a performance gain of over 17% compared to the second-best baselines. This is mainly attributed to the adaptivity of the learnable spectral filters used in DGSDA to graphs with different homophily.

**Traffic Network.** The experimental results on the airport traffic network datasets are shown in Tab. 3. It can be observed that the proposed DGSDA outperforms the baselines on domain adaptation tasks between Brazil and USA, as well as between Europe and USA, which highlights the superiority of DGSDA. While DGSDA demonstrates strong performance in many scenarios, the domain adaptation task between Brazil and Europe presents unique challenges. It can be partially attributed to the limited scale of both do-

mains, resulting in the model overly adapting to the dominant hub and failing to generalize to the whole structure. Anyway, DGSDA still performs better than vanilla GNN-based methods and methods only using node embedding (*i.e.*, UDAGCN, and DANE).

### 5.3. Effectiveness Study

To validate the Bernstein polynomial-based topological alignment mechanism, the frequency response curves of the source and target domain filters are visualized as shown in Fig. 2. The x-axis denotes normalized graph signal frequency $\lambda$ ($\lambda$=0 for low-frequency, $\lambda$=2 for high-frequency components), and the y-axis represents filter gain h($\lambda$). Red/blue colors indicate target/source domains, with solid lines denoting filters jointly optimized under our method and dashed lines showing single-domain baselines.

From Fig. 2, the following three key observations can be made: (1) Independently trained single-domain filters (dashed lines) exhibit consistent trend patterns, revealing structural similarities in underlying spectral distributions across domains. This provides the prerequisite for feasible spectral alignment. (2) The jointly optimized filters (solid lines) demonstrate nearly identical response curves, verifying our method's capability to align cross-domain spectral distributions. (3) Optimized filter responses lie between single-domain baselines while being closer to the target distribution. This suggests that our model preserves information from the source domain while adaptively adjusting to target spectral characteristics, achieving bidirectional alignment rather than unilateral transfer.

### 5.4. Ablation Study

This experiment aims to evaluate the contribution of each constraint. In Fig. 3, $Base$ denotes the variant model using only source cross-entropy loss $L_{source}$. $DGSDA - A$ and $DGSDA - AM$ represent the variant model that sequentially increases $\mathcal{L}_{align}$ and $\mathcal{L}_{mmd}$, respectively. It can be

*Table 3.* Node classification performance on traffic network. The metric is mean accuracy (%) and standard deviation. The best and the second best results are highlighted in bold and underlined, respectively.

| METHOD | B → U | U → B | E → U | U → E | B → E | E → B |
|---|---|---|---|---|---|---|
| GCN | 43.28±3.18 | 49.49±0.48 | 45.01±0.98 | 47.32±0.89 | 44.21±2.05 | 55.88±3.39 |
| GAT | 42.69±5.98 | 50.81±3.87 | 40.92±4.48 | 38.10±2.21 | 34.79±2.29 | 42.14±4.56 |
| DANE | 41.78±1.29 | 40.44±1.01 | 32.38±2.36 | 33.87±0.29 | 33.03±0.29 | 41.98±0.28 |
| UDAGCN | 34.49±2.18 | 36.78±2.86 | 41.26±0.78 | 41.06±0.18 | 43.23±1.72 | 42.33±1.81 |
| ADAGCN | 44.04±3.18 | 55.32±3.18 | 47.89±3.18 | 46.89±2.36 | 50.06±1.21 | 60.28±3.28 |
| STRURW | 52.01±3.21 | 60.01±0.69 | 48.89±3.67 | **52.31±1.08** | 53.23±0.13 | 62.10±1.27 |
| JHGDA | 51.46±3.85 | 60.74±3.24 | 49.03±2.96 | 50.83±3.36 | **54.39±5.76** | **70.12±6.78** |
| KBL | 36.69±1.63 | 35.27±0.75 | 44.28±0.63 | 45.36±0.29 | 46.89±0.96 | 53.03±0.34 |
| PAIRALIGN | 41.76±1.93 | 57.86±2.13 | 44.38±0.69 | 45.68±1.09 | 44.89±0.18 | 53.13±0.21 |
| A2GNN | 51.18±1.38 | 54.47±2.67 | 47.63±2.18 | 50.47±1.96 | 50.33±3.20 | 59.98±1.93 |
| DGSDA | **53.18±0.81** | **61.07±1.60** | **49.73±0.20** | 51.23±0.19 | 49.92±0.96 | 61.37±4.33 |

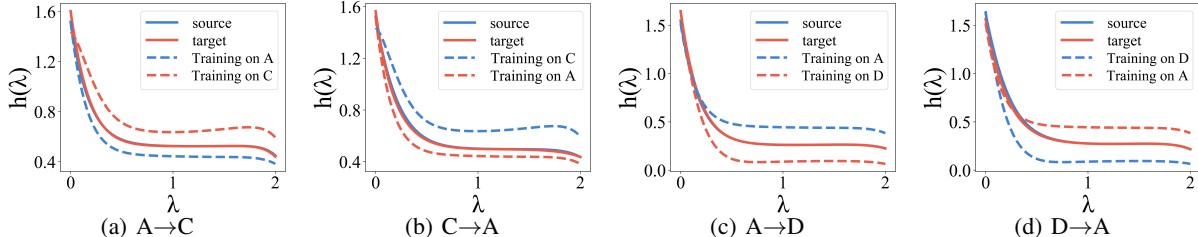

*Figure 2.* Spectral alignment verification via Bernstein polynomial filters. Red/blue colors indicate target/source domains, with solid lines denoting filters jointly optimized under DGSDA and dashed lines showing single-domain baselines.

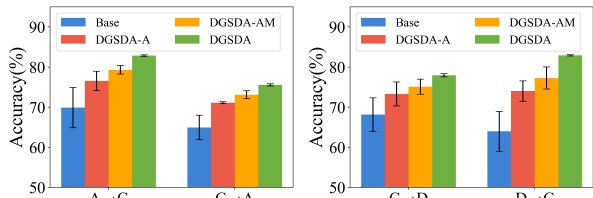

*Figure 3.* Ablation studies on citation networks.

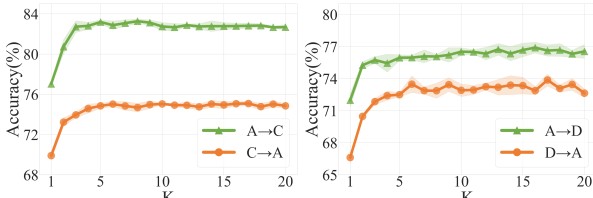

*Figure 4.* The sensitivity of polynomial order K.

observed that as additional constraints are incorporated, the performance of the models steadily improves. This trend underscores that the enhanced performance is a result of the synergistic effect of all the constraints. Moreover, it is particularly noteworthy that the introduction of the proposed model alignment loss significantly boosts the model's performance, thereby highlighting its effectiveness.

### 5.5. Hyperparameter Analysis

These experiments are performed to offer an intuitive understanding for selecting hyper-parameters (including polynomial order $K$ and attribute alignment weight $\beta$).

**Polynomial Order $K$.** It can be observed from Fig. 4 that DGSDA exhibits stable performances with respect to the hyperparameter $K$ in the range $K \geq 5$, with variations remaining within $2\%$. This indicates that the learnable spectral filter has expressive spectral patterns at this stage, enabling it to effectively adapt to these domain transformations. Moreover, this illustrates that DGSDA is not sensitive to

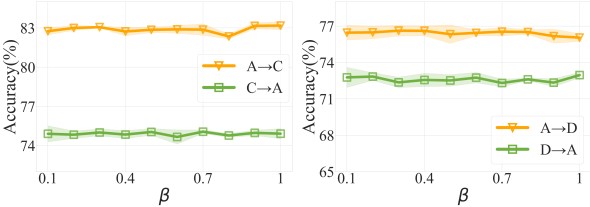

*Figure 5.* The sensitivity of attribute alignment weight $\beta$.

this hyperparameter $K$.

**Attribute alignment weight $\beta$.** From the observations in Fig. 5, it is clear that DGSDA achieve approximately consistent performances across a range of parameter choices. This suggests that DGSDA is not sensitive to the parameter $\beta$. Therefore, we need not pay excessive attention to the specific values of the above two parameters. See Appendix C for a full analysis of the remaining hyperparameters.

# 6. Conclusions

Unsupervised Graph Domain Adaptation (UGDA) faces inherent challenges due to entangled attribute-topology distribution shifts and reliance on fragile pseudo-labels. This work presents Disentangled Graph Spectral Domain Adaptation (DGSDA), which addresses these limitations through spectral filter alignment and distribution shift decoupling. By disentangling attribute embeddings from topology via Bernstein polynomial-based spectral filters, DGSDA circumvents error-prone topology alignment and pseudo-label estimation. The theoretical analysis further establishes that regularizing polynomial coefficients tightens the domain adaptation bound by constraining the Lipschitz continuity of spectral filters. Extensive experiments on various graphs demonstrate the superior performance of DGSDA.

# Impact Statement

This paper presents work whose goal is to advance the field of Machine learning. There are many potential societal consequences of our work, none of which we feel must be specifically highlighted here.

# Acknowledgments

This work was supported in part by the National Natural Science Foundation of China (No. U22B2036, 62376088, 62272020, 62025604, 92370111, 62272340, 62261136549), in part by the Hebei Natural Science Foundation (No. F2024202047), in part by the National Science Fund for Distinguished Young Scholarship (No. 62025602), in part by the Hebei Yanzhao Golden Platform Talent Gathering Programme Core Talent Project (Education Platform) (HJZD202509), in part by the Post-graduate's Innovation Fund Project of Hebei Province (CXZZBS2025036), in part by the Tencent Foundation, and in part by the XPLORER PRIZE.

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

## A. Objective Function Details.

In this section, detailed description about the objective function in Eq. (7) are provided. Each of the loss terms in the overall objective function focuses on different objectives.

**Source domain classification losses.** The $\mathcal{L}_{source}$ term constitutes the fundamental supervised learning component of our framework. Formulated as a cross-entropy loss over labeled source domain data, it directly optimizes the model's prediction accuracy through:

$$\mathcal{L}_{source} = -\frac{1}{N^S} \sum_{i=1}^{N^S} \sum_{c=1}^{C} y_{i,c} \log p_{i,c} \tag{11}$$

**Spectral alignment losses.** $\mathcal{L}_{align}$ focuses on aligning spectral coefficients between the source and target domains, which is provided in Eq. (6).

**Maximum mean discrepancy loss.** Maximum mean discrepancy (MMD) is widely-used for non-graph DA (Gretton et al., 2012), aiming to align feature representations to reduce distribution differences. Here is the specific formula.

$$\mathcal{L}_{mmd} = \frac{1}{(N^S)^2} \sum_{i=1}^{N^S} \sum_{j=1}^{N^S} k\left(H_i^S, H_j^S\right) + \frac{1}{(N^T)^2} \sum_{i=1}^{N^T} \sum_{j=1}^{N^T} k\left(H_i^T, H_j^T\right) - \frac{2}{N^S N^T} \sum_{i=1}^{N^S} \sum_{j=1}^{N^T} k\left(H_i^S, H_j^T\right) \tag{12}$$

**Target domain unsupervised loss.** It promotes model adaptation to the target domain through unsupervised learning:

$$\mathcal{L}_{target} = -\frac{1}{N^T} \sum_{i=1}^{N^T} \hat{y}_i \log \hat{y}_i \tag{13}$$

where $\hat{y}_i$ denotes the predicted labels in target domain.

## B. Proofs for Theorems

### B.1. Proof for Theorem 4.3

**Theorem 4.3** if $f \in \text{Lip}_A \mu$, then for all $n \geq 1$, $B_n(f) \in \text{Lip}_A \mu$ also.

*Proof.* Let $x_1 \leq x_2$ be any two points of $[0,1]$. We need to show that

$$|B_n(f; x_2) - B_n(f; x_1)| \leqslant A(x_2 - x_1)^\mu,$$

given that $f$ satisfies Eq. (8). From Eq. (4),

$$
\begin{aligned}
B_n(f; x_2) &= \sum_{j=0}^{n} \binom{n}{j} (1-x_2)^{n-j} f\left(\frac{j}{n}\right) (x_1 + (x_2 - x_1))^j \\
&= \sum_{j=0}^{n} \binom{n}{j} (1-x_2)^{n-j} f\left(\frac{j}{n}\right) \left\{ \sum_{k=0}^{j} \binom{j}{k} x_1^k (x_2 - x_1)^{j-k} \right\} \\
&= \sum_{j=0}^{n} \sum_{k=0}^{j} \frac{n! x_1^k (x_2 - x_1)^{j-k} (1-x_2)^{n-j}}{k!(j-k)!(n-j)!} f\left(\frac{j}{n}\right)
\end{aligned}
$$

On inverting the order of summation and writing $k + l = j$, then

$$B_n(f; x_2) = \sum_{k=0}^{n} \sum_{l=0}^{n-k} \frac{n!}{k!l!(n-k-l)!} x_1^k (x_2 - x_1)^l \times (1-x_2)^{n-k-l} f\left(\frac{k+l}{n}\right). \tag{14}$$

We now construct a similar double sum for $B_n(f; x_1)$. Again, from Eq. (4), we have

$$
\begin{aligned}
B_n(f; x_1) &= \sum_{k=0}^{n} \binom{n}{k} x_1^k f\left(\frac{k}{n}\right) \left((x_2 - x_1) + (1 - x_2)\right)^{n-k} \\
&= \sum_{k=0}^{n} \binom{n}{k} x_1^k f\left(\frac{k}{n}\right) \left\{\sum_{l=0}^{n-k} \binom{n-k}{l} (x_2 - x_1)^l (1 - x_2)^{n-k}\right\} \\
&= \sum_{k=0}^{n} \sum_{l=0}^{n-k} \frac{n!}{k! l! (n-k-l)!} x_1^k (x_2 - x_1)^l \\
&\quad \times (1 - x_2)^{n-k-l} f\left(\frac{k}{n}\right)
\end{aligned}
\tag{15}
$$

On subtracting Eq. (15) from Eq. (14), we have

$$
\begin{aligned}
&\left| B_n(f; x_2) - B_n(f; x_1) \right| \\
&= \left| \sum_{k=0}^{n} \sum_{l=0}^{n-k} \frac{n!}{k! l! (n-k-l)!} x_1^k (x_2 - x_1)^l (1 - x_2)^{n-k-l} \right. \\
&\quad \left. \times \left\{ f\left(\frac{k+l}{n}\right) - f\left(\frac{k}{n}\right) \right\} \right| \\
&\leqslant A \sum_{k=0}^{n} \sum_{l=0}^{n-k} \frac{n!}{k! l! (n-k-l)!} x_1^k (x_2 - x_1)^l (1 - x_2)^{n-k-l} \left(\frac{l}{n}\right)^{\mu}
\end{aligned}
$$

on using Eq. (8),

$$
\begin{aligned}
&= A \sum_{l=0}^{n} \frac{(x_2 - x_1)^l n!}{l! (n-l)!} \left(\frac{l}{n}\right)^{\mu} \left\{\sum_{k=0}^{n-l} \binom{n-l}{k} x_1^k (1 - x_2)^{n-k-l}\right\} \\
&= A \sum_{l=0}^{n} \binom{n}{l} (x_2 - x_1)^l \left(\frac{l}{n}\right)^{\mu} (x_1 + 1 - x_2)^{n-l} \\
&= A B_n (x^{\mu}; x_2 - x_1), \quad \text{by } Eq.(4), \\
&\leqslant A (x_2 - x_1)^{\mu}
\end{aligned}
$$

Thus we see that $B_n(f) \in \mathrm{Lip}_A \mu$, where A is the Lipschitz constant of $f$ so that the theorem is proved.

## B.2. Proof for Theorem 4.4

**Theorem 4.4.** Suppose the Bernstein polynomial $h_K$ approximate the ground truth one $\bar{f}$ with $\theta_k = \bar{f}(\frac{k}{K})$ and $\|\mathbf{X}\|_{\mathrm{op}} \leq 1$ and $\|\mathbf{W}\|_{\mathrm{op}} \leq 1$ where $\|\cdot\|_{\mathrm{op}}$ stands for operator norm, the following inequality holds:

$$
\begin{aligned}
\|g(G_1) - g(G_2)\|_2 &\leq C_\lambda \left(1 + \tau\sqrt{N}\right) \|\mathbf{A}_1 - \mathbf{P}^* \mathbf{A}_2 \mathbf{P}^{* \top}\|_{\mathrm{F}} \\
&\quad + \mathcal{O}\left(\|\mathbf{A}_1 - \mathbf{P}^* \mathbf{A}_2 \mathbf{P}^{* \top}\|_{\mathrm{F}}^2\right) + \max\{|h_K(\mathbf{\Lambda}_2)|\} \|\mathbf{X}_1 - \mathbf{P}^* \mathbf{X}_2\|_{\mathrm{F}},
\end{aligned}
$$

where $\tau = (\|\mathbf{U}_1 - \mathbf{U}_2\|_{\mathrm{F}} + 1)^2 - 1$ stands for the eigenvector misalignment which can be bounded, $\mathbf{P}^* = \mathrm{argmin}_{\mathbf{P} \in \Pi} \left\{\|\mathbf{X}_1 - \mathbf{P}\mathbf{X}_2\|_{\mathrm{F}} + \|\mathbf{A}_1 - \mathbf{P}\mathbf{A}_2\mathbf{P}^{* \top}\|_{\mathrm{F}}\right\}$, $\Pi$ is the set of permutation matrices, $\mathcal{O}\left(\|\mathbf{A}_1 - \mathbf{P}^* \mathbf{A}_2 \mathbf{P}^{* \top}\|_{\mathrm{F}}^2\right)$ is the remainder term with bounded multipliers, and $C_\lambda$ is the Lipschitz constant of $\bar{f}$ that $\forall \lambda_i, \lambda_j, |\bar{f}(\lambda_i) - \bar{f}(\lambda_j)| \leq C_\lambda |\lambda_i - \lambda_j|$.

*Proof.* Denote the optimal permutation matrix for $G_1, G_2$ as $\mathbf{P}^*$, we compute the difference of the GNN outputs:

$$
\begin{aligned}
& \left\| g\left(G_1\right) - g\left(G_2\right) \right\|_2 \\
={}& \left\| \sigma\left(B_K\left(\mathbf{A}_1\right)\mathbf{X}_1\mathbf{W}\right) - \sigma\left(B_K\left(\mathbf{A}_2\right)\mathbf{X}_2\mathbf{W}\right) \right\|_2 \\
\stackrel{(a)}{=}{}& \left\| \sigma\left(B_K\left(\mathbf{A}_1\right)\mathbf{X}_1\mathbf{W}\right) - \sigma\left(B_K\left(\mathbf{P}^*\mathbf{A}_2\mathbf{P}^{*\mathsf{T}}\right)\mathbf{P}^*\mathbf{X}_2\mathbf{W}\right) \right\|_2 \\
\stackrel{(b)}{\leq}{}& \left\| B_K\left(\mathbf{A}_1\right)\mathbf{X}_1\mathbf{W} - B_K\left(\mathbf{P}^*\mathbf{A}_2\mathbf{P}^{*\mathsf{T}}\right)\mathbf{P}^*\mathbf{X}_2\mathbf{W} \right\|_{\mathsf{F}} \\
\stackrel{(c)}{\leq}{}& \left\| \mathbf{W} \right\|_{\mathrm{op}} \left( \left\| B_K\left(\mathbf{A}_1\right)\mathbf{X}_1 - B_K\left(\mathbf{P}^*\mathbf{A}_2\mathbf{P}^{*\mathsf{T}}\right)\mathbf{X}_1 + B_K\left(\mathbf{P}^*\mathbf{A}_2\mathbf{P}^{*\mathsf{T}}\right)\mathbf{X}_1 - B_K\left(\mathbf{P}^*\mathbf{A}_2\mathbf{P}^{*\mathsf{T}}\right)\mathbf{P}^*\mathbf{X}_2 \right\|_{\mathsf{F}} \right) \\
\stackrel{(d)}{\leq}{}& \left\| \mathbf{W} \right\|_{\mathrm{op}} \left\| \mathbf{X}_1 \right\|_{\mathrm{op}} \left\| B_K\left(\mathbf{A}_1\right) - B_K\left(\mathbf{P}^*\mathbf{A}_2\mathbf{P}^{*\mathsf{T}}\right) \right\|_{\mathsf{F}} + \left\| \mathbf{W} \right\|_{\mathrm{op}} \left\| B_K\left(\mathbf{P}^*\mathbf{A}_2\mathbf{P}^{*\,\mathsf{T}}\right) \right\|_{\mathrm{op}} \left\| \mathbf{X}_1 - \mathbf{P}^*\mathbf{X}_2 \right\|_{\mathsf{F}} \\
\stackrel{(e)}{\leq}{}& \left\| B_K\left(\mathbf{A}_1\right) - B_K\left(\mathbf{P}^*\mathbf{A}_2\mathbf{P}^{*\,\mathsf{T}}\right) \right\|_{\mathsf{F}} + \max\left(\left|h_K\left(\mathbf{\Lambda_2}\right)\right|\right) \left\| \mathbf{X}_1 - \mathbf{P}^*\mathbf{X}_2 \right\|_{\mathsf{F}} \\
\stackrel{(f)}{\leq}{}& C_\lambda\left(1 + \tau\sqrt{N}\right) \left\| \mathbf{A}_1 - \mathbf{P}^*\mathbf{A}_2\mathbf{P}^{*\mathsf{T}} \right\|_{\mathsf{F}} + \mathcal{O}\left(\left\| \mathbf{A}_1 - \mathbf{P}^*\mathbf{A}_2\mathbf{P}^{*\mathsf{T}} \right\|_{\mathsf{F}}^2\right) + \max\left(\left|h_K\left(\mathbf{\Lambda_2}\right)\right|\right) \left\| \mathbf{X}_1 - \mathbf{P}^*\mathbf{X}_2 \right\|_{\mathsf{F}},
\end{aligned}
$$

where (a) is due to the permutation invariance property of graph filters; (b) is achieved with the triangle inequality and the assumption $|\sigma(b) - \sigma(a)| \leq |b - a|, \forall a, b \in \mathbb{R}$; (c) and (d) use the fact that for any two matrices $\mathbf{A}, \mathbf{B}$, $\|\mathbf{AB}\|_{\mathsf{F}} \leq \min(\|\mathbf{A}\|_{\mathrm{op}}\|\mathbf{B}\|_{\mathsf{F}}, \|\mathbf{A}\|_{\mathsf{F}}\|\mathbf{B}\|_{\mathrm{op}})$, and (c) further applies the triangle inequality; (e) adopts the assumption $\|\mathbf{X}\|_{\mathrm{op}} \leq 1$ and $\|\mathbf{W}\|_{\mathrm{op}} \leq 1$ which in practice can be guaranteed with normalization, and easily extended to the case with $\|\mathbf{X}\|_{\mathrm{op}} \leq K, \|\mathbf{W}\|_{\mathrm{op}} \leq K, \forall K > 0$, and because $B_K(\mathbf{P}^*\mathbf{A}_2\mathbf{P}^{*\mathsf{T}}) = (\mathbf{P}^*\mathbf{U}_2)h_K(\mathbf{\Lambda}_2)(\mathbf{P}^*\mathbf{U}_2)^{\mathsf{T}}$ can be diagonalized, its operator norms equal the spectral radius; (f) is the direct outcome borrowed from (Gama et al., 2020) Theorem 1. In common case, $C_\lambda$ is the Lipschitz constant of $B_K$ that $\forall \lambda_i, \lambda_j, |B_K(\lambda_i) - B_K(\lambda_j)| \leq C_\lambda |\lambda_i - \lambda_j|$. According to Theorem 4.3, Bernstein polynomial possesses the same Lipschitz constant as the function to be approximated, i.e., $\bar{f}$. Therefore, the $C_\lambda$ is the Lipschitz constant of $\bar{f}$ that $\bar{f}$ that $\forall \lambda_i, \lambda_j, |\bar{f}(\lambda_i) - \bar{f}(\lambda_j)| \leq C_\lambda |\lambda_i - \lambda_j|$. The proof is completed. $\square$

## B.3. Proof for Theorem 4.5

**Theorem 4.5.** Let's define the matching distance between $G_1, G_2$ as $\eta(G_1, G_2) = \min_{\mathbf{P} \in \Pi}\left\{\|\mathbf{X}_1 - \mathbf{P}\mathbf{X}_2\|_{\mathsf{F}} + \|\mathbf{A}_1 - \mathbf{P}\mathbf{A}_2\mathbf{P}^{*\,\mathsf{T}}\|_{\mathsf{F}}\right\}$. Suppose that the edge perturbation is bounded that $\forall G_1, G_2, \|\mathbf{A}_1 - \mathbf{P}^*\mathbf{A}_2\mathbf{P}^{*\,\mathsf{T}}\|_{\mathsf{F}} \leq \varepsilon$ with the optimal permutation $\mathbf{P}^*$, and there exists an eigenvalue $\lambda^* \in \mathbb{R}$ to achieve the maximum $|h_K(\lambda^*)| < \infty$. The Lipschitz constant of the proposed DGSDA can be estimated as

$$
C_f = \max\left\{C_\lambda K_1 + \varepsilon K_2, |h_K(\lambda^*)|\right\},
$$

where $K_1, K_2$ is the supremes of $\left(1 + \tau\sqrt{N}\right)$ and the remainder multiplier in Theorem 4.4.

*Proof.* To calculate the Lipschitz constant $C_f$ w.r.t the matching distance, based upon Lemma 1, we assure the following inequality:

$$
\begin{aligned}
\left\| g\left(G_1\right) - g\left(G_2\right) \right\|_2 &\leq C_\lambda\left(1 + \tau\sqrt{N_G}\right)\left\| \mathbf{A}_1 - \mathbf{P}^*\mathbf{A}_2\mathbf{P}^{*\mathsf{T}} \right\|_{\mathsf{F}} + \mathcal{O}\left(\left\| \mathbf{A}_1 - \mathbf{P}^*\mathbf{A}_2\mathbf{P}^{*\mathsf{T}} \right\|_{\mathsf{F}}^2\right) + |h_K(\lambda^*)|\left\| \mathbf{X}_1 - \mathbf{P}^*\mathbf{X}_2 \right\|_{\mathsf{F}}, \\
&\leq C_f\eta\left(G_1, G_2\right),
\end{aligned}
$$

the latter inequality of which can be rewritten as:

$$
\begin{aligned}
\left(C_\lambda\left(1 + \tau\sqrt{N}\right)\left\| \mathbf{A}_1 - \mathbf{P}^*\mathbf{A}_2\mathbf{P}^{*\mathsf{T}} \right\|_{\mathsf{F}} + \mathcal{O}\left(\left\| \mathbf{A}_1 - \mathbf{P}^*\mathbf{A}_2\mathbf{P}^{*\mathsf{T}} \right\|_{\mathsf{F}}^2\right) - C_f\left\| \mathbf{A}_1 - \mathbf{P}^*\mathbf{A}_2\mathbf{P}^{*\mathsf{T}} \right\|_{\mathsf{F}}\right) \\
+ \left(|h_K(\lambda^*)| - C_f\right)\left\| \mathbf{X}_1 - \mathbf{P}^*\mathbf{X}_2 \right\|_{\mathsf{F}} \leq 0
\end{aligned}
$$

which is necessary for:

$$
\begin{aligned}
C_\lambda\left(1 + \tau\sqrt{N}\right)\left\| \mathbf{A}_1 - \mathbf{P}^*\mathbf{A}_2\mathbf{P}^{*\mathsf{T}} \right\|_{\mathsf{F}} + \mathcal{O}\left(\left\| \mathbf{A}_1 - \mathbf{P}^*\mathbf{A}_2\mathbf{P}^{*\mathsf{T}} \right\|_{\mathsf{F}}^2\right) - C_f\left\| \mathbf{A}_1 - \mathbf{P}^*\mathbf{A}_2\mathbf{P}^{*\mathsf{T}} \right\|_{\mathsf{F}} &\leq 0 \\
\left(|h_K(\lambda^*)| - C_f\right)\left\| \mathbf{X}_1 - \mathbf{P}^*\mathbf{X}_2 \right\|_{\mathsf{F}} &\leq 0
\end{aligned}
$$

which is equivalent to:

$$C_f \geq C_\lambda K_1 + \varepsilon K_2$$
$$C_f \geq |h_K(\lambda^*)|$$

The bounding of $K_1, K_2$ follows (Gama et al., 2020) Theorem 1 and the first minimum solution can be calculated from the quadratic function w.r.t. the edge matching distance $\left\|\mathbf{A}_1 - \mathbf{P}^*\mathbf{A}_2\mathbf{P}^{*\mathsf{T}}\right\|_{\mathsf{F}}$. Let $C_f$ takes the larger value between them, we complete the proof. $\square$

## C. Experiments Details.

### C.1. Dataset Details

In this section, detailed description about the datasets used in our experiments are provided.

Table 4. Dataset Statistics.

| DATASET | #NODES | #EDGES | #LABELS | #HOMO |
|---|---|---|---|---|
| ACMV9 | 9360 | 31112 | 5 | 0.7998 |
| CITATIONV1 | 8935 | 30196 | 5 | 0.8598 |
| DBLPV7 | 5484 | 16234 | 5 | 0.8189 |
| BLOG1 | 2300 | 66942 | 6 | 0.3991 |
| BLOG2 | 2896 | 107672 | 6 | 0.4002 |
| ENGLAND | 7126 | 35324 | 2 | 0.5560 |
| GERMANY | 9498 | 153138 | 2 | 0.6322 |
| BRAZIL | 131 | 2148 | 4 | 0.4683 |
| EUROPE | 399 | 11990 | 4 | 0.4048 |
| USA | 1190 | 27198 | 4 | 0.6978 |

**ArnetMiner** These datasets are three citation networks obtain from ArnetMiner (Dai et al., 2022): ACMv9 (A), Citationv1 (C), DBLPv7 (D). Nodes are papers, while edges represent citations between papers. The objective is to classify all papers into five distinct research domains: Artificial Intelligence, Computer Vision, Databases, Information Security, and Networking.

**Blog** These datasets are derived from the BlogCatalog dataset (Shen et al., 2020). In these datasets, each node symbolizes a blogger, while the edges denote the friendships among bloggers. The objective is to forecast the group affiliations of these bloggers.

**Twitch** These datasets are Twitch gamer networks from six regions (Liu et al., 2024a): Germany (DE), England (EN), Spain (ES), France (FR), Portugal (PT), and Russia (RU). Nodes are users, while connections signify friendships among them. In this situation, users are divided into two groups depending on whether they use explicit language. Among these datasets, we pay more attention to the two largest datasts, Germany dataset (DE) and England datasets (EN).

**Airport** These datasets are airport traffic networks from three countries (Ribeiro et al., 2017): Brazil (B), Europe (E) and USA (U). Within these datasets, nodes stand for airports, and edges indicate flight links between the airports. The labels classify airports based on their activity levels, measured in the number of flights or passengers.

### C.2. Detailed Hyperparameters.

#### C.2.1. HYPERPARAMETER $\alpha$ AND $\gamma$

This section provides additional insights into hyperparameters $\alpha$ and $\gamma$. As shown in Fig. 6, $\alpha$ demonstrates strong robustness with less than 1% fluctuation across weight initialization schemes. For $\gamma$, empirical studies reveal 0.05 achieves optimal performance, so experiments consequently adopt this fixed value.

#### C.2.2. HYPERPARAMETER $K$

This section extends the analysis of K (Bernstein polynomial order) to low-homophily social networks. Experimental results reveal two patterns: (1) Heterophilic graphs require substantially larger K (e.g., K≥15 vs. K≥5 for homophilic

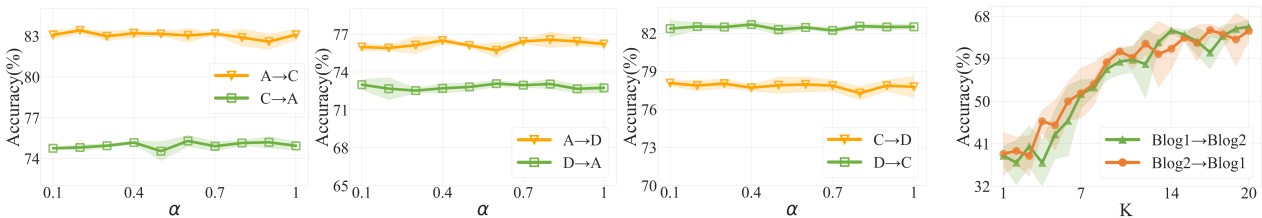

| Figure 6. The sensitivity of $\alpha$. | Figure 7. The sensitivity of K. |

graphs) to reach saturation accuracy, potentially due to complex neighborhood patterns demanding higher-order filtering; (2) Performance stabilizes when K≥15, indicating sufficient signal modeling capacity at this threshold.

# D. More Experiments.

### D.1. Pseudo-labels verification.

This section verifies the feasibility of using predicted pseudo-labels on the target domain. A variant model named DGSDA+PL is introduced, which combines pseudo-labels of the target domain. The compared results reveal that pseudo-labels consistently lead to performance degradation in all domain adaptation scenarios, demonstrating the infeasibility of the mentioned scheme.

Table 5. Pseudo-labels experiment on citation network. The metric is mean accuracy (%) and standard deviation.

|           | A → C        | C → A        | A → D        | D → A        | C → D        | D → C        |
|-----------|--------------|--------------|--------------|--------------|--------------|--------------|
| DGSDA     | 83.57±0.22   | 75.54±0.28   | 76.90±0.51   | 74.07±0.56   | 78.38±0.28   | 82.92±0.15   |
| DGSDA+PL  | 81.23±2.52   | 74.40±2.22   | 75.36±2.37   | 71.36±1.33   | 77.03±1.04   | 79.45±1.49   |

This is primarily due to the low reliability of the pseudo-labels generated in the early stages of training, which can cause error accumulation in learning processes, and the noise amplification effect in graph neural networks, where erroneous pseudo-labels propagate through message-passing mechanisms. This is also the reason why the proposed method outperforms topology alignment with pseudo-labels.

### D.2. Disentanglement effectiveness validation.

To clarify the sources of performance improvement, this experiment introduced a variant of DGSDA that uses Bernstein polynomials and directly aligns node representations instead of separately aligning attributes and topology. As demonstrated in Tab. 6, this variant consistently underperforms compared to the full DGSDA model, which suggests that the explicit disentanglement of attributes and topology plays a crucial role in enhancing model effectiveness.

Table 6. Disentanglement experiment on citation network. The metric is mean accuracy (%) and standard deviation.

|               | A → C        | C → A        | A → D        | D → A        | C → D        | D → C        |
|---------------|--------------|--------------|--------------|--------------|--------------|--------------|
| DGSDA         | 83.57±0.22   | 75.54±0.28   | 76.90±0.51   | 74.07±0.56   | 78.38±0.28   | 82.92±0.15   |
| VARIANT MODEL | 81.01±0.32   | 73.25±0.22   | 73.15±0.19   | 72.03±0.24   | 76.32±0.17   | 80.25±0.21   |

### D.3. Topology pattern capture capability study.

This section verifies that DGSDA is able to capture the correct topological patterns even in unsupervised scenarios. In the experiment, the supervised variant of DGSDA employs the supervised loss from 10% labeled data in the target domain, replacing the unsupervised loss: the spectral alignment loss and the entropy loss. The results are shown in Tab. 6.

The results indicate that the unsupervised DGSDA achieves comparable performance to the supervised version, highlighting the effectiveness of the unsupervised losses. This can be attributed to two key factors. First, by regularizing the coefficients of Bernstein polynomials (in Eq. (4)), the method explicitly aligns the spectral filters across different domains. This alignment enables the target filters to inherit topology-aware patterns from the source domain, even without labels. Second, the entropy

*Table 7.* Topology pattern capture capability experiment on citation network. The metric is mean accuracy (%) and standard deviation.

| | A → C | C → A | A → D | D → A | C → D | D → C |
|---|---|---|---|---|---|---|
| DGSDA | 83.57±0.22 | 75.54±0.28 | 76.90±0.51 | 74.07±0.56 | 78.38±0.28 | 82.92±0.15 |
| DGSDA(SUPERVISED) | 83.20±0.52 | 76.37±2.75 | 79.49±0.56 | 77.10±0.83 | 80.16±0.65 | 83.03±0.48 |

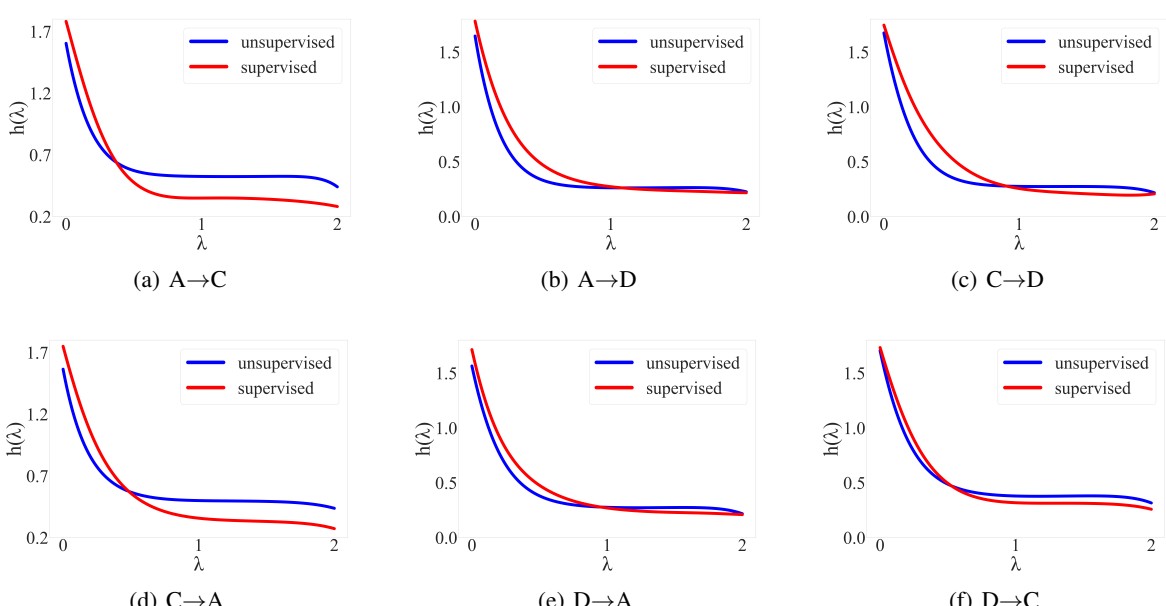

*Figure 8.* Comparison of filter curves in unsupervised and supervised scenarios.

loss sharpens cluster assignments, which implicitly encourages the model to learn more discriminative topological features.

Moreover, from the observations in Fig. 8, the learned filter curves are similar in both scenarios, which also suggests that the target domain parameters capture consistent topological patterns in the unsupervised scenario as well.

