# OpenReview forum: "Disentangled Graph Spectral Domain Adaptation"
_ICML.cc/2025/Conference — ICML 2025 poster_

### Official Review · Reviewer_fWcK · 2025-03-05

**Overall Recommendation:** 4

**Summary:**

To break away from the attribute and topology entanglement on Unsupervised Domain Adaptation (UDA), this paper introduces a novel method, DGSDA, directly aligning complicated graph spectral filters. This paper conducts experiments on various types of graph datasets to demonstrate the effectiveness of DGSDA.

**Claims And Evidence:**

Yes, the claims made in the submission are supported by clear and convincing evidence from both experimental and theoretical aspects.

**Essential References Not Discussed:**

No, there are no essential related works missing in the paper that need further discussion.

**Experimental Designs Or Analyses:**

Yes, I have checked the soundness of the experimental designs (including the compared methods and experimental setups) and analyses.

**Methods And Evaluation Criteria:**

The proposed DGSDA is well aligned with the problem of UGDA. The disentanglement of attribute and topology alignments, the use of spectral filter alignment, and the comprehensive experiments on diverse datasets collectively demonstrate its effectiveness.

**Other Comments Or Suggestions:**

1) Some equations, such as Eq. (4) and Eq. (9), as well as Theorem 4.4, appear to be slightly misaligned. It is recommended to adjust the line breaks for better formatting.

2) On line 182: It possesses the following **three** advantages instead of **two**.

3) The name "PairAlign" is misspelled as "PariAlign" in Tables 1 and 2.

**Other Strengths And Weaknesses:**

**Strengths**

1) The paper is well-written and easy to follow.
2) Experimental results on benchmark datasets are provided.

**Weaknesses**

The authors fail to clearly attribute the source of performance improvement in their method. First, DGSDA employs Bernstein polynomials, which are not commonly used in comparison methods. This choice alone may contribute to the performance gains, making it unclear how much of the improvement is due to the disentanglement strategy itself. Further empirical results are needed to isolate the specific role of disentanglement. Additionally, given the lack of labels in the target domain, it is unclear how the model ensures that the target domain parameters accurately capture the topological patterns.

**Questions For Authors:**

Refer to Weaknesses.

**Relation To Broader Scientific Literature:**

The key contributions are directly related to the broader literature by proposing disentanglement techniques to solve the problems in traditional UGDA. It leverages recent advancements in spectral GNNs and builds on theoretical foundations of Lipschitz continuity and model alignment.

**Theoretical Claims:**

Yes, I have checked the correctness of the proofs for the theoretical claims presented in the paper. Specifically, I have verified the proofs for Theorems 4.3, 4.4, and 4.5, which are central to the theoretical analysis of DGSDA.

---

> ### Author Rebuttal · Authors · 2025-04-01
>
> > Q1. The authors fail to clearly attribute the source of performance improvement in their method. Further empirical results are needed to isolate the specific role of disentanglement.
>
> R1. To address your concerns, we have conducted an additional experiment to clearly identify the source of performance improvement. This experiment introduced a variant of DGSDA that uses Bernstein polynomials and directly aligns node representations instead of separately aligning attributes and topology. The variant model's performance is consistently worse than that of our full DGSDA, as shown in the following table. This indicates that the disentanglement play a crucial role in enhancing the performance of our model.
>
> |     | A→C | C→A | A→D | D→A | C→D | D→C |
> | --- | --- | --- | --- | --- | --- | --- |
> | DGSDA | 83.57$\pm$0.22 | 75.54$\pm$0.28 | 76.90$\pm$0.51 | 74.07$\pm$0.56 | 78.38$\pm$0.28 | 82.92$\pm$0.15 |
> | variant model | 81.01$\pm$0.32 | 73.25$\pm$0.22 | 73.15$\pm$0.19 | 72.03$\pm$0.24 | 76.32$\pm$0.17 | 80.25$\pm$0.21 |
>
> ---
>
> > Q2. Given the lack of labels in the target domain, it is unclear how the model ensures that the target domain parameters accurately capture the topological patterns.
>
> R2. We have conducted two additional experiments to demonstrate that the unsupervised loss can provide effects similar to supervised loss in terms of model parameter optimization. In the experiment, the supervised variant of DGSDA employs the supervised loss from 10% labeled data in the target domain, replacing the unsupervised loss: the spectral alignment loss and the entropy loss. The results are shown below.
>
> |     | A→C | C→A | A→D | D→A | C→D | D→C |
> | --- | --- | --- | --- | --- | --- | --- |
> | DGSDA | 83.57$\pm$0.22 | 75.54$\pm$0.28 | 76.90$\pm$0.51 | 74.07$\pm$0.56 | 78.38$\pm$0.28 | 82.92$\pm$0.15 |
> | DGSDA (supervised) | 83.20$\pm$0.52 | 76.37$\pm$2.75 | 79.49$\pm$0.56 | 77.10$\pm$0.83 | 80.16$\pm$0.65 | 83.03$\pm$0.48 |
>
> The results indicate that the unsupervised DGSDA achieves comparable performance to the supervised version, highlighting the effectiveness of the unsupervised losses. This can be attributed to two key factors. First, by regularizing the coefficients of Bernstein polynomials (in Eq. 6), the method explicitly aligns the spectral filters across different domains. This alignment enables the target filters to inherit topology-aware patterns from the source domain, even without labels. Second, the entropy loss sharpens cluster assignments, which implicitly encourages the model to learn more discriminative topological features.
>
> In addition, we have compared the learned filter curves in both labeled and unlabeled target domains. The results can be found at https://anonymous.4open.science/r/DGSDA/figure/DC.png. In the supervised learning case, the model parameters capture the homophily topological pattern, characterized by increasing low-frequency information and suppressing high-frequency information. Similarly, in the unsupervised learning case, the target domain parameters can capture the same topological pattern.
>
> ---
>
> > Q3. Some equations, such as Eq. (4) and Eq. (9), as well as Theorem 4.4, appear to be slightly misaligned. It is recommended to adjust the line breaks for better formatting.
>
> R3. We will adjust the line breaks in Eq. (4), Eq. (9), and Theorem 4.4 to ensure proper alignment and improve the overall formatting.
>
> ---
>
> > Q4. Expression error on line 182 and spelling mistake of model "PairAlign".
>
> R4. We will perform a thorough review of the manuscript to correct any expression errors and spelling mistakes.

---

> > ### Comment · Reviewer_fWcK · 2025-04-03
> >
> > The authors' rebuttal addresses all my concerns. After checking the comments from other reviewers, I raise my score.

---

### Official Review · Reviewer_Fybq · 2025-03-06

**Overall Recommendation:** 4

**Summary:**

This study addresses the challenge of unsupervised graph domain adaptation in scenarios involving distribution shifts and missing labels by proposing a novel solution that disentangles the distribution shift. Specifically, the method DGSDA refines the topology alignment into GNN alignment and incorporates spectral filter alignment loss.

**Claims And Evidence:**

This paper provides a comprehensive evaluation of the DGSDA model, supported by both theoretical analysis and extensive experiments, thereby effectively demonstrating its efficacy. Thus, the claims made in the paper are well-substantiated by clear and convincing evidence.

**Essential References Not Discussed:**

No other related works that are essential to understanding the (context for) key contributions need to be discussed or cited.

**Experimental Designs Or Analyses:**

I examined all the experimental designs and analyses in Section 5 and Section B, and believe it effectively demonstrates the properties of the proposed model.

**Methods And Evaluation Criteria:**

The proposed method involves (1) disentangling embedding alignment into topology and attribute alignments and (2) exploiting alignments of filter parameters to flexibly implement topology alignment, both of which make sense for the graph-domain adaptation problem.

**Other Comments Or Suggestions:**

1) In Section 4.1 Distribution Shift Disentanglement, “Topology alignment中“the graph data shift can be simplified from $P^S(A, X|Y) \neq P^T (A, X|Y)$ to $P^S(A|X, Y) = P^T (A|X, Y)$” seems to be misstatement. The correct statement should be $P^S(A|X, Y) \neq P^T (A|X, Y)$.

**Other Strengths And Weaknesses:**

**Strengths**

1）The idea of GNN alignment is interesting.
2）The method is simple yet has solid theoretical support.

**Weaknesses**

1) The title appears to be somewhat ambiguous. The title does not reflect the focus on the **unsupervised** problem in graph domain adaptation, which is a key aspect of the study.

2) The notation $T$ used in the paper is not clear. For example, $\mathbf{X}^{T}$ represents the node attribute matrix of the target domain, but it can also be interpreted as the transpose of the node attribute matrix.

3) The description of the proposed method lacks clarity. While $L_{source}$ and $L_{mmd}$ are described in words, they are not accompanied by clear, formal formulations.

**Questions For Authors:**

See Weaknesses.

**Relation To Broader Scientific Literature:**

Current graph domain adaption works focus on proposing topology alignment strategies, including aligning the edge distributions of two domains using the CSBM. DGSDA utilizes a new filter alignment to improve flexibility.

**Theoretical Claims:**

After a detailed examination of the proof, I have essentially confirmed its correctness.

---

> ### Author Rebuttal · Authors · 2025-04-01
>
> > Q1. The title appears to be somewhat ambiguous. The title does not reflect the focus on the **unsupervised** problem in graph domain adaptation, which is a key aspect of the study.
>
> R1. Thank you for pointing this out. The primary focus of this paper is indeed on the unsupervised problem in graph domain adaptation. While our architecture can also accommodate target labels when available, the unsupervised scenario remains our main emphasis. We will consider revising the title to better reflect this focus.
>
> ---
>
> > Q2. The notation T used in the paper is not clear. For example, X^T represents the node attribute matrix of the target domain, but it can also be interpreted as the transpose of the node attribute matrix.
>
> R2. Thanks for your careful check. We will correct $X^T$ to $X^{\top}$ to denote the transpose of the matrix.
>
> ---
>
> > Q3. The description of the proposed method lacks clarity. While $L_{source}$ and $L_{mmd}$ are described in words, they are not accompanied by clear, formal formulations.
>
> R3. The formal formulations of $L_{source}$ and $L_{mmd}$ are presented as follows:
>
> $\mathcal{L}\_{source} = -\frac{1}{N^{{S}}} \sum\_{i=1}^{N^{{S}}} \sum\_{c=1}^{C} y\_{i,c} \log p\_{i,c}$
>
> $\mathcal{L}\_{mmd} = \frac{1}{\left(N^{{S}}\right)^2} \sum\_{i=1}^{N^{{S}}} \sum\_{j=1}^{N^{{S}}} k\left(H\_i^{{S}}, H\_j^{{S}}\right) + \frac{1}{\left(N^{{T}}\right)^2} \sum\_{i=1}^{N^{T}} \sum\_{j=1}^{N^{T}} k\left(H\_i^{T}, H\_j^{T}\right) - \frac{2}{N^{{S}} N^{T}} \sum\_{i=1}^{N^{{S}}} \sum\_{j=1}^{N^{T}} k\left(H\_i^{{S}}, H\_j^{T}\right) $
>
> where $k(·,·)$ represents the kernel function.
>
> We will add them to the appendix to enhance the clarity of the manuscript.
>
> ---
>
> > Q4. In Section 4.1 Distribution Shift Disentanglement, "Topology alignment in the graph data shift can be simplified from $P^{S}(A,X| Y) \neq P^{T}(A,X| Y)$ to $P^{S}(A| X,Y) = P^{T}(A| X,Y)$" seems to be misstatement. The correct statement should be $P^{S}(A| X,Y) \neq P^{T}(A| X,Y)$.
>
> R4. Thanks for pointing out this important detail. We will correct this formula in the revised manuscript and ensure that all related discussions are consistent with this accurate representation.

---

### Official Review · Reviewer_SNRK · 2025-03-10

**Overall Recommendation:** 4

**Summary:**

This paper introduces a novel pipeline for unsupervised graph domain adaptation by disentangling attribute and topology alignments by considering that attribute alignment has been widely investigated. Based on the aligned node attribute, the topology alignment is converted to the model alignment by taking into consideration the widely developed GNN models. Then, the Bernstein polynomial is employed as the backbone for its approximation property and spectral perspective. Theoretical analysis and experimental evaluations justify the pipeline and proposed models.

**Claims And Evidence:**

The correctness of the introduced pipeline is verified by the derivation from the Bayesian theorem. The replacement of topology alignment with model alignment makes sense due to the connection between topology and GNN models. Theoretical analysis and experiments demonstrate the statements.

**Essential References Not Discussed:**

Sufficient. It covers most existing competitive SOTA and important milestones.

**Experimental Designs Or Analyses:**

The experiments are extensive, including quantitative and qualitative analyses. The setting is widely used in this field, and the baselines are recently proposed competitive ones. Thus, the experimental evaluations are convincing.

**Methods And Evaluation Criteria:**

As shown in the previous section, I think both the pipeline and the proposed GNN alignment make sense. Besides, the employment of these two strategies reduces the requirement of the pseudo label in the topology alignment, which often relies on the accurate estimation of the node membership.

**Other Comments Or Suggestions:**

None

**Other Strengths And Weaknesses:**

This paper possesses high originality, which may inspire the following variants on model alignment. The claims are justified with a rigorous theory investigation and extensive experiments. The main weakness is the lack of source code. Since the proposed method is flexible and complicated with four terms as the objective function, it is necessary to provide source code to make the read easy to get the implementation details.

**Questions For Authors:**

Although the authors claim the use of model alignment avoids the requirements of pseudo-labels, I wonder whether pseudo-labels also benefit the model alignment since the polynomial coefficients can also be learned from labels as supervision. Whether the predicted pseudo-labels on the target domain can be employed?

The theoretical analysis focuses on the Bernstein spectral GNN alignment. Does the decomposition strategy possess solid theoretical findings?

Unsupervised graph domain adaptation is often composed of multiple terms as the objective function, and thus, training it is difficult to balance the impacts of terms. Can these terms be unified to felicitate the training?

**Relation To Broader Scientific Literature:**

Unsupervised graph domain adaptation is a critical topic in graph learning for the graph foundation model design. Although there exist GNN-based methods, as reviewed in the related work section,  this paper gives a novel methodology by both decomposition and model parameter alignment. This is more powerful and efficient compared to the existing ones by considering the spectral perspective.

**Theoretical Claims:**

I cursorily examined the proof of the theorems in the appendix and believe they are correct.

---

> ### Author Rebuttal · Authors · 2025-04-01
>
> > Q1. The main weakness is the lack of source code.
>
> R1. The source code has been made available at (https://anonymous.4open.science/r/DGSDA) for verification purposes. We promise to make the code public once this paper is accepted.
>
> ---
>
> > Q2. Whether the predicted pseudo-labels on the target domain can be employed?
>
> R2. To answer your valuable question, we have conducted experiments to verify the feasibility of using predicted pseudo-labels on the target domain. This experiment introduces a variant model named DGSDA+PL, which combines pseudo-labels of the target domain. The compared results reveal that pseudo-labels consistently lead to performance degradation in all domain adaptation scenarios, demonstrating the infeasibility of the mentioned scheme.
>
> |     | A→C | C→A | A→D | D→A | C→D | D→C |
> | --- | --- | --- | --- | --- | --- | --- |
> | DGSDA | 83.57$\pm$0.22 | 75.54$\pm$0.28 | 76.90$\pm$0.51 | 74.07$\pm$0.56 | 78.38$\pm$0.28 | 82.92$\pm$0.15 |
> | DGSDA+PL | 81.23$\pm$2.52 | 74.40$\pm$2.22 | 75.36$\pm$2.37 | 71.16$\pm$1.33 | 77.03$\pm$1.04 | 79.45$\pm$1.49 |
>
> This is primarily due to the low reliability of the pseudo-labels generated in the early stages of training, which can cause error accumulation in learning processes, and the noise amplification effect in graph neural networks, where erroneous pseudo-labels propagate through message-passing mechanisms. This is also the reason why the proposed method outperforms topology alignment with pseudo-labels.
>
> ---
>
> > Q3. Does the decomposition strategy possess solid theoretical findings?
>
> R3. We acknowledge that the current analysis focuses on demonstrating the **feasibility** of disentanglement without providing a precise error bound between the entangled and disentangled representations. This is a common limitation in the graph disentanglement field, where theoretical guarantees are still lacking. We will strive to address this in future work.
>
> ---
>
> > Q4. Unsupervised graph domain adaptation is often composed of multiple terms as the objective function, and thus, training it is difficult to balance the impacts of terms. Can these terms be unified to felicitate the training?
>
> R4. We understand your concern for the stability of training. Unfortunately, these terms cannot be unified, as each of the loss terms focuses on different objectives as other GDA methods. To be specific, $L_{source}$ targets minimizing prediction error in the source domain, ensuring effective training on labeled source data. $L_{align}$ focuses on aligning spectral coefficients between the source and target domains. $L_{mmd} $aims to align feature representations to reduce distribution differences. $L_{target}$ promotes model adaptation to the target domain through unsupervised learning. Moreover, the experiments in the hyper-parameter analysis demonstrated that our model is relatively robust to the hyper-parameters used for weighting these terms. We will explore integrating these loss terms in future work to facilitate easier balancing.

---

### Official Review · Reviewer_er9T · 2025-03-11

**Overall Recommendation:** 4

**Summary:**

This paper proposes Disentangled Graph Spectral Domain Adaptation (DGSDA) to alleviate the inaccuracies of pseudo-labels and the limited expressive ability of graph encoders to capture rich topology information. It decomposes the attribute and topology alignments and replaces the topology alignment with the powerful model alignment. To harness the parameter efficiency of spectral GNNs, the Bernstein polynomial is employed, and the polynomial coefficients are aligned. Theoretical analysis shows its rationality and superiority compared to existing ones. Experimental experiments also justify the claims.

**Claims And Evidence:**

The rationality and superiority of the proposed DGSDA are supported by both theoretical and experimental evidence. It is clear and convincing.

**Essential References Not Discussed:**

The references are sufficient.

**Experimental Designs Or Analyses:**

The soundness of the experiments is checked. The design is based on widely-employed datasets and criteria. The performances are verified on varying datasets. The ablation study and hyper-parameter analysis are conducted.

**Methods And Evaluation Criteria:**

The proposed DGSDA makes sense and is novel by decomposing attribute and topology alignments. The experimental evaluations are reasonable with widely-sed criteria.

**Other Comments Or Suggestions:**

1. Some explanations should clarify the theoretical differences from [You et al., 2023].
2. It is better to provide the source code to enhance reproducibility.

**Other Strengths And Weaknesses:**

**Strengths**

The motivations are interesting and make sense.

The proposed method is novel and solid.

The theoretical justification is rigorous.

The experimental evaluations are convincing.

**Weakness**

The symbols, especially the theory part, are too complex to read.

**Questions For Authors:**

1. Why is the performance of the proposed method lower than that of JHGDA on the traffic network dataset?

2. The legend in Figure 2 is not clear. What are the differences between source/target and A/C.

3. Why are the results in Figure 2 continuous curves? I think they should be discrete values.

**Relation To Broader Scientific Literature:**

UGDA is an important topic in the graph machine learning field. Previous work focuses on the employment of DA methods in i.i.d. data. This paper is along the line of topology alignment. It alleviates the issue of pseudo-label inaccuracy by adopting spectral model alignment beyond the topology one. Therefore, it is novel. Besides, the decomposition of topology and attribute alignment are also novel and interesting.

**Theoretical Claims:**

The correctness of theorems is checked, as well as their proofs in the appendix. However, the symbols are very complex.

---

> ### Author Rebuttal · Authors · 2025-03-31
>
> > Q1. The symbols, especially the theory part, are too complex to read.
>
> R1. Thanks for your feedback. We will thoroughly review and modify all the symbols to make them easier to read.
>
> ---
>
> > Q2. Some explanations should clarify the theoretical differences from [You et al., 2023].
>
> R2. The theoretical differences between this paper and the mentioned work [You et al., 2023] is  **Polynomial Choice & Lipschitz Properties**: The proposed DGSDA adopts Bernstein polynomials, whose Lipschitz constant $C_{\lambda}$ is determined by the ground-truth function (in Theorem 4.3), rather than being restricted by the basic polynomial coefficients as in the work [You et al., 2023]. This allows more flexible and accurate spectral domain adaptation.
>
> ---
>
> > Q3. It is better to provide the source code to enhance reproducibility.
>
> R3. According to your suggestion, the source code has been made available at (https://anonymous.4open.science/r/DGSDA) for verification purposes.
>
> ---
>
> > Q4. Why is the performance of the proposed method lower than that of JHGDA on the traffic network dataset?
>
> R4. The proposed method generally outperforms JHGDA on most tasks in the traffic network dataset and is less effective than JHGDA on the $B → E$ and $E → B$ tasks. The performance weakness may be attributed to overfitting due to limited training data. The Brazil and Europe datasets contain a relatively small number of nodes, which makes the proposed models with multiple constraints more prone to over-capturing the patterns of individual hub nodes. This, in turn, makes it difficult to effectively generalize to the overall structure of the target domain. Nonetheless, the extensive results illustrate the effectiveness of the proposed method.
>
> ---
>
> > Q5. The legend in Figure 2 is not clear. What are the differences between source/target and A/C.
>
> R5.  The solid lines represent the filter curves trained on the domain adaptation tasks, while the dashed lines represent the filter curves obtained by training BernNet only on the corresponding datasets. Thus, “source” and “target” in the legend mean the filter curves of source domain encoder and target domain encoder trained on the domain adaptation tasks; “A” and “C” in the legend denote the filter curves of BernNet on the A and C datasets, respectively. We will change them to “Training on A” or “Training on C” in the revised manuscript to enhance readability.
>
> ---
>
> > Q6. Why are the results in Figure 2 continuous curves?
>
> R6. Figure 2 shows the learned polynomials instead of their coefficients, and thus contain continuous curves. The x-axis in Figure 2 denotes the normalized graph signal frequency $\lambda$, and the y-axis represents filter gain $h(\lambda)$. They are independent of the polynomial order $K$ and coefficients $\theta_k$. The Bernstein polynomial is defined as $h(\lambda)=\sum_{k=0}^K\theta_kb^K_k(\frac{\lambda}{2})$, where $b^K_k(t)$ is the Bernstein basis function. For any given $\lambda$, it is first mapped to the [0, 1] interval (i.e., $\frac{\lambda}{2}$), and $h(\lambda)$ is calculated using the learned coefficients $\theta_k$ and the corresponding Bernstein basis functions $b^K_k$. Thus, by sampling sufficiently many $\lambda$ values, we can plot the continuous curves shown in Figure 2.

---

### Decision · Program_Chairs · 2025-05-01

**Decision:**

Accept (poster)

**Comment:**

**Summary:** This paper proposes DGSDA, a new method for unsupervised graph domain adaptation (UGDA) that disentangles attribute and topology alignment and introduces spectral filter alignment via Bernstein polynomials. The approach avoids reliance on pseudo-labels for topology modeling and instead aligns model parameters directly in the spectral domain. The paper presents theoretical insights (including a bound and analysis of Lipschitz continuity) and extensive experiments across multiple benchmark datasets, demonstrating the effectiveness of DGSDA. The rebuttal included further clarifications, new ablation results, and additional supervised comparisons.

**Decision:** All reviewers gave accept recommendations and highlighted the novelty, technical soundness, and practical value of the proposed approach. The method offers a fresh perspective by replacing traditional topology alignment with spectral model alignment, and the use of Bernstein polynomials is well-motivated both theoretically and empirically. While some concerns were raised—such as equation formatting, title clarity, and disentanglement attribution—the authors addressed them comprehensively in the rebuttal. In particular, they provided empirical comparisons to isolate the role of disentanglement and demonstrated that the unsupervised objective is competitive with a supervised variant. Overall, the paper is well-written, the contributions are meaningful, and the empirical validation is thorough. Therefore, I recommend acceptance.